

# Simulation of SOA Formation from the Photooxidation of Monoalkylbenzenes in the Presence of Aqueous Aerosols Containing Electrolytes under Various NO$_x$ Levels

Chufan Zhou, Myoseon Jang, and Zechen Yu

Department of Environmental Engineering Sciences, University of Florida, Gainesville, 32611, USA

*Correspondence to*: Myoseon Jang (mjang@ufl.edu)

**Abstract.** The formation of secondary organic aerosols (SOAs) from the photooxidation of three monoalkylbenzenes (toluene, ethylbenzene, and n-propylbenzene) in the presence of inorganic seeds (SO$_4^{2-}$-NH$_4^+$-H$_2$O system) under varying NO$_x$ levels

has been simulated using the Unified Partitioning-Aerosol Phase Reaction (UNIPAR) model. The evolution of the volatility-reactivity distribution (mass-base stoichiometric coefficient, $\alpha_i$) of oxygenated products, which were created by the near-explicit gas kinetic mechanism, was integrated with the model using the parameters linked to the concentrations of HO$_2$ and RO$_2$ radicals. This dynamic distribution was applied to estimate the model parameters related to the thermodynamic constants of the products in multiple phases (e.g., the gas phase, organic phase, and inorganic phase) and the reaction rate constants in

the aerosol phase. The SOA mass was predicted through the partitioning and aerosol chemistry processes of the oxygenated products in both the organic phase and aqueous solution containing electrolytes, with the assumption of organic-inorganic phase separation. The prediction of the time series SOA mass (12-hr), against the aerosol data obtained from an outdoor photochemical smog chamber, was improved by the dynamic $\alpha_i$ set compared to the prediction using the fixed $\alpha_i$ set. Overall, the effect of an aqueous phase containing electrolytes on SOA yields was more important than that of the NO$_x$ level under our

simulated conditions or the utilization of the age-driven $\alpha_i$ set. Regardless of the NO$_x$ conditions, the SOA yields for the three aromatics were significantly higher in the presence of wet electrolytic seeds than those obtained with dry seeds or no seed. When increasing the NO$_x$ level, the fraction of organic matter (OM) produced by aqueous reactions to the total OM increased due to the increased formation of relatively volatile organic nitrates and peroxyacyl nitrate like products. The predicted partitioning mass fraction increased as the alkyl chain length increases but the organic mass produced via aerosol phase

reactions decreased due to the increased activity coefficient of the organic compounds containing longer alkyl chains. Overall, the lower mass-base SOA yield was seen in the longer alkyl-substituted benzene in both the presence and absence of inorganic seeded aerosols. However, the difference of mole-base SOA yields of three monoalkylbenzenes becomes small because the highly reactive organic species (i.e., glyoxal) mainly originates from ring opening products without alkyl side chain. UNIPAR predicted the conversion of hydrophilic, acidic sulfur species to non-electrolytic dialkyl-organosulfate (diOS) in the aerosol.





Thus, the model predicted the impact of diOS on both hygroscopicity and acidity, which subsequently influenced aerosol growth via aqueous reactions.

# 1 Introduction

Anthropogenic volatile organic compounds (VOCs) have significant impacts on urban and regional atmospheric chemistry, despite fewer global emissions (McDonald et al., 2018). As an important group of anthropogenic VOCs, aromatic hydrocarbons (HCs) are emitted from automobile exhaust (Zhang et al., 2018) and solvent use (Cheng et al., 2018) and are known to be precursors for secondary organic aerosols (SOAs), which are formed during the process of photooxidation (Seinfeld and Pandis, 2016). In polluted areas (e.g., urban areas in Asia), aromatic HCs occupy 11 % to 25 % of the total nonmethane HC emissions (67.0 Tg in 2010) (Li et al., 2017) and traditionally comprise approximately 15 % of SOA formation (Ait-Helal et al., 2014), which contributes to the urban budget of fine particulate matter (Wood et al., 2010).

SOA formation has attracted substantial interest from scholars because of its vital role in affecting regional weather (IPCC, 2015;Seinfeld and Pandis, 2016), urban visibility (Chen et al., 2012;Ren et al., 2018), and health (Requia et al., 2018). The prediction of SOA formation was first fulfilled by a gas-particle partitioning model. The partitioning-based SOA model uses two surrogate products (Odum et al., 1996) or several semivolatile surrogates (e.g., volatility basis set (VBS)) (Donahue et al., 2006), with semiempirical parameters (e.g., the product stoichiometric coefficient ($\alpha$) and gas-particle partitioning coefficient ($K_p$)) for each HC system under a given $NO_x$ condition. Due to its simplicity and high efficiency, the partitioning-base model has been widely used in regional and global models. Nonetheless, the models and their predecessors are limited to predict SOAs formed from in-particle chemistry due to the loss of product structures, which govern the reactivity of organic species in the aerosol phase. Overall, regional air quality models have historically underestimated fine particulate matter in summertime (Appel et al., 2017;Huang et al., 2017) due to the lack of in-particle chemistry, particularly in the presence of an aqueous phase containing electrolytes (Ervens et al., 2011;Tsigaridis et al., 2014;Kelly et al., 2018).

A few models have attempted to implement in-particle chemistry into SOA models. For instance, Johnson et al. (2004) (2005) simulated aromatic SOA chamber data, with a modified $K_p$, to obtain experimentally comparable results, while the delayed simulated SOA mass indicated the occurrence of chemical reactions in the aerosol phase. McNeill et al. (2012) developed the Gas Aerosol Model for Mechanism Analysis (GAMMA) to predict the formation of SOAs via aqueous phase chemistry, which was further applied to the production of isoprene SOAs. Im et al. (2014) advanced the Unified Partitioning-Aerosol Phase Reaction (UNIPAR) model, which predicted the SOA mass from partitioning processes and aerosol-phase reactions (reactions in both organic and inorganic phases and organosulfate (OS) formation). In that study, toluene and 1,3,5-trimethylbenzene SOAs were modeled using near-explicit products with the organic-inorganic phase separation mode. Beardsley and Jang (2016) extended UNIPAR to simulate isoprene SOAs in the single homogeneously mixed phase (organic-inorganic mixture). Despite



the reasonable prediction of SOA masses, UNIPAR faced inaccuracies in predicting time series SOA data due to the use of a fixed (nonage-driven) mass-based stoichiometric coefficient ($\alpha_i$) set.

Age-driven functionalization and fragmentation alter the volatility and reactivity of products and their molecular structures (Donahue et al., 2006;Rudich et al., 2007;Shilling et al., 2007;Hartikainen et al., 2018) which, in turn, varies the in-particle chemistry. Cappa and Wilson (2012) employed tunable parameters to kinetically demonstrate the evolution of SOA mass and the bulk oxygen-to-carbon atomic ratio (O:C ratio) during photochemical aging. However, oligomerization reactions in the aerosol phase were excluded. Donahue et al. (2011) developed a 2D-VBS method, which represented product aging by remapping the volatility and polarity (O:C ratio) of the products in 2D space. Zhao et al. (2015) reported a discrepancy in the simulated toluene SOAs and $\alpha$-pinene SOAs within the same 2D-VBS configuration, which may result from the different reactivities of the oxidation products of the precursors in aerosol-phase reactions. In this study, we have attempted to improve the UNIPAR model by using dynamic (age-driven) $\alpha_i$ and applying the resulting model to predict the SOA formation of three monoalkylbenzenes (i.e., toluene, ethylbenzene, and n-propylbenzene) under a wide range of environmental conditions (i.e., $NO_x$, temperature, humidity, sunlight and aerosol acidity). To consider the effect of the aging process on SOA formation, model parameters related to the organic molecular structures (i.e., the molecular weight (MW) and O:C ratio) and the $\alpha_i$ set are calculated as the system ages, allowing for the internally dynamic estimation of the activity coefficient of the products (lumping species) in the aqueous phase containing electrolytes. Hence, the model is able to dynamically compute the partitioning coefficient of organics in the inorganic phase ($K_{in}$) by reflecting the photochemical evolution of the products in the gas phase and, consequently, improving SOA prediction. Organosulfate (OS), which has been identified in both laboratory and field studies (Hettiyadura et al., 2015;Li et al., 2016a;Estillore et al., 2016;Chen et al., 2018), is an important chemical species due to its low volatility and ability to modulate the hygroscopicity of sulfate constituents. In the presence of acidic sulfate constituents, UNIPAR also predicts the production of non-electrolytic sulfates (i.e., dialkyl-substituted OS (diOS)) and the ensuing modification of aqueous phase reactions. The feasibility of unified rate constants for aerosol-phase reactions was evaluated by extending the preexisting rate constants, which has been employed for toluene and 1,3,5-trimethylbenzene (Im et al., 2014) and isoprene (Beardsley and Jang, 2016), to the three monoalkylbenzenes in this study.

## 2 Experimental techniques

The SOA formation from the photooxidation of monoalkylbenzenes were conducted in the University of Florida Atmospheric PHotochemical Outdoor Reactor (UF-APHOR) (Table 1). The concentrations of HCs, trace gasses ($NO_x$, $SO_2$, and $O_3$), inorganic ions, aerosol acidity and organic carbon (OC) of particles were monitored, as were the meteorological factors (i.e., relative humidity (RH), temperature, and ultraviolet (UV) radiation). The configurations of the chamber and instrumentations were described by Im et al. (2014), Li et al. (2016a), Beardsley and Jang (2016), Yu et al. (2017), and Jiang et al. (2017). Aerosol acidity ([$H^+$], mol/L of aerosol) is monitored using colorimetry integrated with the reflectance UV-visible spectrometer



(C-RUV) technique (Li et al. (2015a)) (Section S4 in the supporting information (SI)). The diOS concentration (μmol m$^{-3}$) in an aerosol is estimated by the difference [H$^+$] obtained from ion chromatography (IC) interfaced with a particle-into-liquid sampler (PILS) (Li et al. (2015a) and C-RUV method. Each HC was studied under at least two NO$_x$ levels (high NO$_x$: HC/NO$_x$ < 5.5; low NO$_x$: HC/NO$_x$ > 5.5) with or without inorganic seeded aerosols (i.e., sulfuric acid (SA) or ammonium sulfate (AS)).

HONO was added into the system as a reaction initiator. To investigate the effect of the liquid water content (LWC) on AS seeded SOA, two RH conditions were applied. (1) dry: RH < efflorescence RH (ERH) of the AS seed; (2) wet: RH > 50 % to prevent crystallization of AS seed. The ratio of organic matter (OM) to OC was experimentally determined to be 1.9 (Table 1, EB4), which was similar to the reported value of 2.0 (Kleindienst et al., 2007).

## 3 Model descriptions

The structure of the UNIPAR model is illustrated in Fig. 1. The simulation of aromatic SOA formation in the aqueous phase containing electrolyte was performed under the assumption of complete organic-inorganic phase separation. Bertram et al. (2011) modeled the separation RH (SRH) in the liquid-liquid phase of the mixture of organic and AS using the bulk O:C ratio. When ambient RH < SRH, the system undergoes organic-inorganic phase separation. The reported O:C ratios of the toluene, ethylbenzene, and n-propylbenzene SOAs were 0.62 (Sato et al., 2012), 0.55 (Sato et al., 2012), and 0.45 (Li et al., 2016b),

respectively, which caused the corresponding SRH values to be 65 %, 80 %, and 93 %, respectively. Most RH for active photooxidation of HCs under ambient sunlight were under 65%, which supported the assumption of organic-inorganic phase separation. In addition, as less soluble oligomers formed in the aerosol phase, an SRH higher than 65 % was more likely to be yielded.

### 3.1 Atmospheric evolution of lumping species

The gas-phase oxidation of HCs is simulated using the near-explicit gas-phase chemistry mechanism (Master Chemical Mechanism (MCM) v 3.3.1) (Jenkin et al., 2012) integrated with the Morpho chemical solver (Jeffries et al., 1998). To represent the polluted urban and clean environments, the gas-phase oxidation is simulated under various NO$_x$ levels (HC ppbC/NO$_x$ ppb=2 - 45) for given meteorological conditions (e.g., sunlight, temperature, and RH on 06/14/18 near the summer solstice, with a clear sky in Gainesville, Florida). The resulting oxygenated products are lumped into 51 species within a 2D

set with 8 levels of volatility (1-8: 10$^{-8}$, 10$^{-6}$, 10$^{-5}$, 10$^{-4}$, 10$^{-3}$, 10$^{-2}$, 10$^{-1}$, and 1 mmHg) and 6 levels of aerosol-phase reactivity (very fast: VF, fast: F, medium: M, slow: S, partitioning only: P, and multi-alcohol: MA) plus 3 additional reactive species (glyoxal (GLY), methylglyoxal (MGLY), and epoxydiols (IEPOX, isoprene products)) with own vapour pressure. The detailed lumping criteria and $\alpha_i$ equations are described in Section S1 in the SI along with the major product structures (Tables S1-S3). To simulate age-dependent SOA formation, $\alpha_i$ is reconstructed over time by a weighted average method using a pair of gas-

phase oxidation compositions with different aging statuses: fresh composition and highly oxidized composition. The weighting factor is related to an aging scale factor ($f_A$), which is defined as





$$f_A = log \frac{[HO_2]+[RO_2]}{[HC]_0}, \tag{1}$$

where $[RO_2]$ and $[HO_2]$ represent the concentrations (ppb) of $RO_2$ and $HO_2$ radicals, respectively, and $[HC]_0$ represents the initial HC concentration (ppbC). The lower boundaries of $f_A$ to determine the $\alpha_i$ set of fresh composition is equal to -7.2 at HC/NO$_x$=2 (high NO$_x$ levels) and -3.7 at HC/NO$_x$=14 (low NO$_x$ levels). The upper value of $\alpha_i$ set for highly aged composition

is obtained when $f_A$ is equal to -5.2 and -2.9 under the same high and low NO$_x$ levels, respectively. Both the fresh $\alpha_i$ and highly aged $\alpha_i$ are functions of HC/NO$_x$. $f_A$ is further converted into a fractional aging factor, $f_A'$, ranging from 0 - 1 at each NO$_x$ level (0: fresh composition; 1: highly aged composition). Then, $\alpha_i$ is dynamically reconstructed based on $f_A'$ under varying NO$_x$ conditions.

$$\alpha_i = (1 - f_A')(\text{fresh } \alpha_i) + (f_A')(\text{highly aged } \alpha_i) \tag{2}$$

The molecular structures, including $O:C_i$ and MW ($MW_i$), of each species ($i$) are also dynamically represented by a similar method, as shown in the SI in Section S2.

### 3.2 SOA formation: partitioning

The partitioning coefficient ($K_P$) from the gas ($g$) phase to the organic ($or$) phase ($K_{or,i}$, m$^3$ µg$^{-1}$) and from the $g$ phase to the inorganic ($in$) phase ($K_{in,i}$, m$^3$ µg$^{-1}$) of each species is estimated using the following gas-particle absorption model (Pankow,

1994).

$$K_{or,i} = \frac{7.501\,R\,T}{10^9\,MW_{or}\,\gamma_{or,i}\,p_{l,i}^0} \quad \text{and} \quad K_{in,i} = \frac{7.501\,R\,T}{10^9\,MW_{in}\,\gamma_{in,i}\,p_{l,i}^0}, \tag{3}$$

where $R$ represents the gas constant (8.314 J mol$^{-1}$ K$^{-1}$). $T$ represents the ambient temperature (K). $MW_{or}$ and $MW_{in}$ represent the average MW (g mol$^{-1}$) of organic and inorganic aerosols, respectively. $p_{l,i}^0$ represents the subcooled liquid vapor pressure (mmHg) of a species, $i$. In the $or$ phase, we assume that the activity coefficient ($\gamma_{or,i}$) of a species ($i$) is unity (Jang and Kamens,

1998). In the $in$ phase, $\gamma_{in,i}$ is semi-empirically predicted by a regression equation, which was fit the theoretical activity coefficients of various organic compounds to RH, fractional sulfate (FS), and molecular structures (i.e., $MW_i$ and $O:C_i$). FS is a numerical indicator for inorganic compositions related to aerosol acidity ($FS = \frac{[SO_4^{2-}]}{[SO_4^{2-}]+[NH_4^+]}$, where $[SO_4^{2-}]$ and $[NH_4^+]$ are the concentration of the total sulfate and the total ammonium, respectively). The theoretical activity coefficients were estimated at a given humidity and an aerosol composition through a thermodynamic model (Aerosol Inorganic-Organic

Mixtures Functional Groups Activity Coefficients (AIOMFAC)) (Zuend et al., 2011).

$$\gamma_{in,i} = e^{-11.136\ln(100 \times RH) - 13.306 FS - 12.667 O:C_i + 0.06555 MW_i + 62.0614}, \tag{4}$$

The statistical information for Eq. 4 is shown in the SI in Section S3. The resulting $K_{or,i}$ and $K_{in,i}$ are employed to calculate the concentration (µg m$^{-3}$) of the lumping species in multiple phases ($C_{g,i}$, $C_{or,i}$, $C_{in,i}$, and $C_{T,i} = C_{g,i} + C_{or,i} + C_{in,i}$).





Schell et al. (2001) developed a partitioning model to predict SOA formation. This model was reconstructed by Cao and Jang (2010) to include OM formed via aerosol-phase reactions ($OM_{AR,i}$) for a species ($i$), which is estimated in Section 3.3. $OM$ formed during the partitioning process ($OM_P$) is estimated by utilizing the mass balance shown in the following equation.

$$OM_P = \sum_{ij}\left[ C_{T,i} - OM_{AR,i} - C_{g,i}^* \frac{\frac{C_{or,i}}{MW_i}}{\sum_{ij}(\frac{C_{or,i}'}{MW_i} + \frac{OM_{AR,i}}{MW_{oli,i}}) + OM_0} \right], \tag{5}$$

$C_{g,i}^*$ ($1/K_{or,i}$) is the effective saturation concentration and $OM_0$ represents the concentration (mol m$^{-3}$) of the preexisting $OM$. $MW_{oli,i}$ represents the average MW of oligomeric products. Eq. 5 is solved via iterations using the globally converging Newton-Raphson method (Press et al., 1992).

### 3.3 SOA formation: aerosol-phase reactions

The formation of $OM_{AR,i}$ is processed in both $or$ and $in$ phases: oligomerization in the $or$ phase to form $OM_{AR,or,i}$ and
oligomerization in the $in$ phase to form $OM_{AR,in,i}$ based on the assumption of a self-dimerization reaction (i.e., second-order reaction) (Odian, 2004) for organic compounds in media. Oligomerization in an aqueous phase can be accelerated under acidic environment (Jang et al., 2002). The oligomerization rate constants (L mol$^{-1}$ s$^{-1}$) in the $or$ phase and $in$ phase are $k_{o,i}$ and $k_{AC,i}$, respectively, and the kinetic equations for oligomerizations are written as follows.

$$\frac{dC_{or,i}}{dt} = -k_{o,i}C_{or,i}'^2\left(\frac{MW_i OM_T}{\rho_{or}\,10^3}\right), \tag{6}$$

$$\frac{dC_{in,i}}{dt} = -k_{AC,i}C_{in,i}'^2\left(\frac{MW_i M_{in}}{\rho_{in}\,10^3}\right), \tag{7}$$

The bracketed terms in the equations indicate the conversion factors from aerosol-base concentrations ($C_{or,i}'$ and $C_{in,i}'$: mol L$^{-1}$) into air-base concentrations (µg m$^{-3}$) (Section S5). $\rho_{or}$ and $\rho_{in}$ represent the density of the aerosol of $or$ and $in$ aerosol. $\rho_{or}$ was experimentally determined (EB4 in Table 1) to be 1.38 g cm$^{-3}$, which was similar to the reported value of 1.4 g cm$^{-3}$ (Nakao et al., 2011;Chen et al., 2017). $\rho_{in}$ is obtained from a regression equation through the extended aerosol inorganic model
(E-AIM)(Clegg et al., 1998). Due to atmospheric diurnal patterns (high RH at nighttime to low humidity during daytime), it is likely that the RH changes would be based on inorganic aerosol ERH. UNIPAR internally predicts the ERH using the equation derived by Colberg et al. (2003).

$k_{AC,i}$ in Eq. 7 is estimated based on a semiempirical model developed by Jang et al. (2005) as a function of species reactivity
($R_i$), protonation equilibrium constant ($pK_{BH^+_i}$), excess acidity ($X$), water activity ($a_w$), and proton concentration ([H$^+$]), which are estimated by the E-AIM.

$$k_{AC,i} = 10^{1.3R_i + 0.0005\,pK_{BH^+_i} + 1.3\cdot X + \log(a_w[H^+]) - 5.5}, \tag{8}$$





In the *or* phase, $k_{o,i}$ is estimated by excluding the $X$ and $a_w$ $[H^+]$ terms. The formed $OM_{AR}$ can be calculated as a sum of $OM_{AR,or,i}$ and $OM_{AR,or,i}$ for each species assuming that $OM_{AR}$ is irreversibility and nonvolatility (Kleindienst et al., 2006;Cao and Jang, 2010).

### 3.4 Organosulfate formation

In the presence of aqueous acidic sulfate, UNIPAR predicts the formation of diOS ($[diOS]_{model}$) to compute the change in aerosol hygroscopicity and acidity. At each time step, free electrolytic sulfate ($[SO_4^{2-}]_{free}$), which is the sulfate that is unassociated with ammonium ($[NH_4^+]$), is represented as ($[SO_4^{2-}] - 0.5 [NH_4^+]$). $[SO_4^{2-}]_{free}$ is then applied to the semiempirical equation tested previously for several SOA systems (Im et al., 2014;Beardsley and Jang, 2016) to predict $[diOS]_{model}$, as described below,

$$\frac{[diOS]_{model}}{[SO_4^{2-}]_{free}} = 1 - \frac{1}{1 + f_{diOS}\frac{N_{diOS}}{[SO_4^{2-}]_{free}}}, \tag{9}$$

where $f_{diOS}$ represents the diOS conversion factor introduced by Im et al. (2014), which was semi-empirically determined to be 0.063 in this study. $N_{diOS}$ represents the numeric parameter for scaling lumping groups based on the effectiveness of the chemical species to form diOS. For example, the diOS scale factor is 1 for each alcohol and aldehyde group and 2 for each epoxide group (see Tables S1-S3 for functional groups). Then, $N_{diOS}$ is summed at each time step and applied to Eq. 9.

### 3.5 Operation of the UNIPAR model

The variables, which include HC consumption (ΔHC), [HO₂], [RO₂], HC/NO$_x$, RH, temperature, and the inorganic concentration (i.e., Δ[SO₄²⁻] and Δ[NH₄⁺]), were input to the UNIPAR model every 6 minutes (Δt=6 minutes).

### 4 Results and discussion

### 4.1 Prediction of SOA mass under the evolution of oxygenated products

As reported in former studies, the kinetic mechanism tends to underestimate the decay of aromatic HCs because of the low prediction of OH radicals (Johnson et al., 2005;Bloss et al., 2005). In this study, the addition of artificial OH radicals varies with the HC/NO$_x$ ratio by fitting the predicted decay of HCs using the kinetic mechanism in the experimental measurements. The time profiles of the decays of the three HCs are shown in Fig. S1 in the SI. When the NO$_x$ level is very low, the maximum additional OH radical production rate for monoalkylbenzenes is 2×10⁸ molecules cm³ s⁻¹, which is less than 4×10⁸ (Bloss et al., 2005) but similar to the value reported by Im et al. (2014). When HC/NO$_x$ < 3, no addition of artificial OH radicals is needed for the chamber simulation of the decay of monoalkylbenzenes. For the make-up OH production rate constants of all three HCs under varying NO$_x$, the mathematical weighting equation is written below,

$$dynamic\ makeup\ OH\ rate = \frac{e^{0.6 \times HC/NOx}}{e^{0.6 \times HC/NOx} + 50} \times 2.0 \times 10^8\ \text{molecules cm}^3\ \text{s}^{-1}, \tag{10}$$





In our model, we assume that the oxidation of products progresses in the gas phase. Lambe et al. (2012) reported that the transition point of n-$C_{10}$ SOAs from a functionalization dominant regime to a fragmentation dominant regime is approximately 3 days (photochemical equivalent age under an atmospheric OH exposure of $1.5 \times 10^6$ molecules $cm^{-3}$). Under this criterion, we exclude the aging of nonvolatile aerosol products ($OM_{AR}$). However, the oxidation of aerosol products for longer periods of

5 time may decrease the volatility (George and Abbatt, 2010;Jimenez et al., 2009).

Fig. 2 illustrates the evolution of the volatility-reactivity-based distribution of the mass-based stoichiometric coefficient ($\alpha_i$) at the two different $NO_x$ levels (HC/$NO_x$=2.9 and 10.5). Collectively, most $\alpha_i$ values at both $NO_x$ levels tend to decline as the reaction time lapses (Fig. 2(a) vs. Fig. 2(b); Fig. 2(c) vs. Fig. 2(d)) since the evolution of some semivolatile organic compounds

(SVOCs) forms very volatile molecules (i.e., $CO_2$, formic acid, and formaldehyde). For example, the $\alpha_i$ values of highly reactive carbonyls with high volatility (GLY and MGLY in Table S1) are high under the fresh condition and significantly decline as the system ages, because they undergo fast photolysis under sunlight (George et al., 2015;Henry and Donahue, 2012). Consequently, the decay of these highly reactive species leads to the decrease in the production of $OM_{AR}$. The high $NO_x$ level delays the oxidation of gas-phase products. Similar trends in the $\alpha_i$ set can be found for ethylbenzene and n-

propylbenzene, as shown in Figs. S2 and S3, respectively. The $\alpha_i$ of highly reactive species (e.g., GLY, 8VF, 3M, and 5S) decreases by increasing the $NO_x$ level due to the suppression of the $HO_x$ cycle via the reaction of $NO_2$ with OH radicals. As seen in Fig. 2(d), some medium reactivity species (i.e., 2-methyl-4-oxo-3-nitro-2-butenoic acid (3M), 2-methyl-4-oxo-2-butenoic acid (6M), and acetyl-3-oxopropanoate (7M)) start to form as $NO_x$ decreased.

In Fig. 3, the comparison between the simulations of SOA formation from toluene oxidation is based on two different $\alpha_i$-reconstruction strategies: dynamic $\alpha_i$ and fixed $\alpha_i$. A clear improvement in the prediction of SOA formation is demonstrated when comparing the SOA mass using dynamic $\alpha_i$ to that using fixed $\alpha_i$. The aged SOA growth from the three systems (i.e., low $NO_x$ level (Fig. 3(a) and Fig. 3(d)), moderate $NO_x$ level (Fig. 3(b) and Fig. 3(e)), and high $NO_x$ level with an inorganic seed (Fig. 3(c) and Fig. 3(f)) are even smaller than that predicted with the less-aged $\alpha_i$ set, which is fixed at the time of the

HCs being consumed half of the total consumption. Our model simulation against the chamber data suggests that while aging may alter aerosol compositions (Fig. 2), it does not always increase SOA yields. Traditionally, the SOA mass has been predicted using fixed thermodynamic parameters (i.e., $K_p$ and $\alpha_i$), which is inadequate when reflecting upon practical scenarios, where oxygenated product distributions vary dynamically with oxidation.

## 4.2 Effects of aerosol acidity and LWC on SOA formation

In the model, aerosol acidity was expressed using a fractional free sulfate (FFS), which is defined as FFS=([$SO_4^{2-}$] - 0.5[$NH_4^+$])/[$SO_4^{2-}$]. Humidity can influence both aerosol acidity and LWC, which are the model parameters in UNIPAR. Thus, UNIPAR has the capability to decouple the effect of aerosol acidity and humidity, as shown in Fig. 4 for toluene SOA. The impact of aerosol acidity and humidity on the yields of SOAs derived from ethylbenzene and n-propylbenzene is illustrated in





Figure S4. The dramatic difference in SOA yields appears between the RH above ERH and the RH below ERH. The LWC disappears below ERH, and there are no aqueous reactions. For example, the observed SOA yield of ethylbenzene with effloresced AS was significantly smaller than that with wet AS: 11 % (EB8 in Table 1) vs. 30 % (EB9 in Table 1). Kamens et al. (2011) and Liu et al. (2018) reported a significantly lower yield of toluene SOA for dry AS seeded aerosols compared with its wet counterpart. The partitioning of polar carbonaceous products increases with increasing LWC and, thus, aqueous reactions. In the presence of wet aerosols, SOA yields gradually increase with increasing FFS (increasing acidity) at a given RH due to acid-catalyzed oligomerization. The oxygenated products of toluene are relatively more polar than those of ethylbenzene or propylbenzene and positively attribute to increase of $OM_{AR}$.

Compared to isoprene SOAs reported by Beardsley and Jang (2016), the impacts of humidity and acidity on the SOA yields of monoalkylbenzenes in this study are relatively weaker above the ERH (Fig. 4), except for the highly acidic conditions under high humidity. The formation of aromatic SOAs is contributed to a few highly reactive species, such as GLY. Other aromatic oxidation products partitioned in the aerosol phase have moderate solubility and they are slow to react in the aqueous phase. Isoprene products are more hygroscopic than aromatic products and even mixable with an aqueous phase containing electrolytes. The reactions of medium reactivity polar products that formed during isoprene oxidation can be accelerated by an acid catalyst with higher sensitivities to acidity and humidity.

### 4.3 Organosulfate: simulation vs. measurement

Fig. 5 illustrates the time profiles of the predicted concentrations of diOS ([diOS]$_{model}$) and protons ([H$^+$]) with the measured concentrations of diOS ([diOS]$_{exp}$), [NH$_4^+$], and [SO$_4^{2-}$] for different aromatic HCs under given experimental conditions (Fig. 5(a)-(c)). For the three SA seeded SOA experiments, the fractions of diOS to the total sulfate amount are 0.09, 0.15, and 0.06 for toluene (Exp. Tol8, HC/NO$_x$=2.9, FS changing from 0.64 to 0.39), ethylbenzene (Exp. EB7, HC/NO$_x$=12.3, FS changing from 0.82 to 0.46), and n-propylbenzene (Exp. PB5, HC/NO$_x$=14.4, FS changing from 0.76 to 0.38), respectively. The [diOS]$_{model}$ reasonably agrees with [diOS]$_{exp}$. The aerosols in Exp. Tol8 and Exp. PB5 show the cease in diOS formation at approximately 10 am since they became effloresced due to the neutralization of SA with ammonia under the reduction in humidity during the daytime. The diOS fraction in Exp. EB7, which contained wet acidic aerosols, was higher than those in Exp. Tol8 and Exp. PB5 indicating that the acidic condition was favorable for the formation of diOS (Surratt et al., 2010;Lin et al., 2013).

Beardsley and Jang (2016) reported that the diOS fraction for isoprene SOAs was 0.26 (HC/NO$_x$=32.5, FS changing from 0.69 to 0.47), which was more than that for toluene SOAs, indicating that the oxidation products of isoprene may contain more reactive species to form diOS. For example, IEPOX products in isoprene SOAs are known to be reactive to SA (Budisulistiorini et al., 2017). Additionally, isoprene aerosol products are mixable with aqueous solutions containing electrolytes, and they can more effectively form diOS compared to the aromatic products in liquid-liquid phase separation.





To estimate the potential upper boundary of the concentration of diOS ([diOS]$_{max}$) in monoalkylbenzene SOA, the aerosol composition was predicted by the model in the presence of SA aerosols (without neutralization with ammonia) under the given experimental conditions shown in Fig. 5. The resulting diOS fractions were 0.29 (OM-to-sulfate mass ratio (OM:sulf) = 1.4),

0.25 (OM:sulf = 1.4), and 0.12 (OM:sulf = 0.7) for toluene, ethylbenzene and n-propylbenzene, respectively. The aerosol acidity of the ambient aerosol is generally lower than ammonium hydrogen sulfate (AHS) and, thus, the diOS fraction in ambient air would be much lower than the estimated upper boundary. Fig. 5 suggests that the change in both aerosol acidity and hygroscopicity by the formation of nonelectrolytic sulfate is important to predict SOA mass.

### 4.4 Effect of NO$_x$ on SOA formation in the presence of an aqueous phase containing electrolytes

Fig. 6 shows the impact of NO$_x$ on the three aromatic SOAs in this study by producing SOAs at two different NO$_x$ levels in the presence and absence of SO$_2$. Overall, regardless of the inorganic seed conditions, both the chamber observation and model simulation suggest that increasing the NO$_x$ level leads to the decreased formation of SOAs. This trend in the absence of inorganic seed aerosols has also been observed multiple times (Li et al., 2015;Ng et al., 2007;Song et al., 2005). By increasing NO$_x$, the path of an RO$_2$ radical progresses to the formation of organonitrate and peroxyacyl nitrate (PAN) products, which are

less reactive to aerosol-phase reactions. They are relatively volatile and, thus, insignificantly attribute to partitioning SOA mass.

For example, the SOA yields under the low NO$_x$ level (HC(ppbC)/NO$_x$(ppb)=9.1~14.8, Table 1) in the presence of SO$_2$, with a similar degree of ammonia titration (i.e., similar FS values by the end of the experiments), were higher than those without

seeded aerosols: 42 % for toluene (Exp. Tol1, FS=0.44), 26 % for ethylbenzene (Exp. EB1, FS=0.37), and 66 % for propylbenzene (Exp. PB1, FS=0.43). The impact of aerosol acidity was even greater for SOAs produced under a high NO$_x$ level (HC(ppbC)/NO$_x$(ppb)=2.8~5.0): 65 % for toluene (Exp. Tol3, FS=0. 43), 146 % for ethylbenzene (Exp. EB5, FS=0. 39), and 77 % for propylbenzene (Exp. PB3, FS=0. 40). SOA formation under high NO$_x$ conditions is generally more sensitive to aerosol acidity compared to that at low NO$_x$ levels (Figs. 6(a)-(c) and Fig. S5(f) vs. Figs. 6(d)-(f)). The fractions of medium

reactivity products are relatively high in high NO$_x$ levels and their reactions in aerosol phase can be accelerated by an acid catalyst.

### 4.5 Sensitivity of SOA formation to humidity, temperature, aerosol acidity, precursor HCs, and NO$_x$ level

Fig. 7 illustrates the sensitivity of the SOA mass simulated at relatively low concentration of HC (20 ppb) (panel I) for three monoalkylbenzenes to important variables (i.e., humidity (A-I vs. B-I for AHS and C-I vs. D-I for AS), temperature (A-I vs.

E-I for AHS and F-I vs. G-I without inorganic aerosol), aerosol acidity (A-I vs. C-I at RH=45 % and B-I vs. D-I at RH=65 %), and NO$_x$ levels (A-I vs. H-I with AHS seeded aerosols and F-I vs. I-I without inorganic seeded aerosols)). The most drastic change appears by changing the temperature from 273 K (E-I) to 298 K (A-I). The SOA yield is known to increase by 20-150





%, which results from a 10 K decrease in temperature (Sheehan and Bowman, 2001). For all SOAs, noticeable changes are shown between the absence (F-I) and presence (A-I) of wet inorganic seeds, while a minor change appears between wet AHS (A-I) and wet AS (C-I). Within the wet acidic aerosols (A-I vs. B-I and C-I vs. D-I), the effect of RH is insignificant, as discussed in Section 4.2. Although the impact of $NO_x$ (A-I vs. H-I and F-I vs. I-I) is less than that of temperature and inorganic seeds, SOA yields are still significantly altered, as discussed in Section 4.4.

Panel II series in Figure 7 illustrates SOA growth curves under various conditions shown in panel I. Overall, the simulated SOA yields (slopes) increase with a decreased alkyl chain length (toluene > ethylbenzene > n-propylbenzene), which is consistent with our chamber observations (Table 1). Although the decreased of vapor pressure of products benefits increases in $OM_P$ as the alkyl chain length increases, the increase of the activity coefficient of the organic products containing longer alkyl chains in aqueous phase is unfavorable to form $OM_{AR}$ via aqueous reactions. However, the difference of mole-base SOA yields of three monoalkylbenzenes becomes small because the highly reactive organic species (i.e., glyoxal), which are produced through ring opening reactions without alkyl side chain, significantly attribute to $OM_{AR}$. Panel II confirms that the effect of an aqueous phase containing electrolytes on SOA yields is more critical than that of the $NO_x$ level under our simulated conditions.

## 4.6 Sensitivity of model prediction to major variables and model uncertainty

To determine the model sensitivity to these parameters, simulations (Exp. Tol9 in Table 1) were performed by increasing/decreasing vapor pressure ($V_p$), the enthalpy of vaporization ($H_{vap}$), $\gamma_{in,i}$, and $k_{AC,i}$ by factors of 1.5, 1.1, 1.5, and 2, respectively. The corresponding changes in the SOA mass are 27.4 %/-20.1 %, 0.2 %/-0.2 %, 15.9 %/-19.3 %, and 17 %/-17 %, respectively. The change in SOA mass from the baseline for each simulation is shown in Fig. S6.

The uncertainty associated with the group contribution method used for $V_p$ estimation is a factor of 1.45 (Zhao et al., 1999). $H_{vap}$ has a reported error of 2.6 % (Kolska et al., 2005). $\gamma_{in,i}$ is estimated as a function of O:C, MW, RH, and FS (Eq. 4). $k_{AC,i}$ is semi-empirically calculated based on [$H^+$], LWC, and species reactivity (Eq. 8). The E-AIM is performed to estimate the LWC, which is reliable and based on a broadly used water activity dataset (Zhang et al., 2000). Yet, the inorganic thermodynamic models including E-AIM performed inadequately in the prediction of [$H^+$] under low RH and ammonia rich conditions (FS < 0.55) (Li and Jang, 2012).

Although most identified toluene products have been included, such as methyl-cyclohexene (3S), 2-methyl-5-nitrophenol (5P), 2-methyl-benzoquione (6S), 2-methyl-4-oxo-2-butenoic acid (6M), o-cresol (7P), 3-hydroxy-1,3-propandial (7VF), 3-methyl-2(5H)-furanone (8P), MGLY, and GLY (Forstner et al., 1997;Jang and Kamens, 2001;Sato et al., 2007;Gomez Alvarez et al., 2007;Huang et al., 2016), a large amount of toluene oxidation mechanisms and involved products remain unstudied. A similar trend can be found in ethylbenzene and propylbenzene. Evidently, the addition of artificial OH radicals in the gas-phase



simulation suggests missing mechanisms in the MCM v 3.3.1 or an improper branching ratio of reactions. Additionally, the diverse reactions of the $RO_2$ radicals might be oversimplified in the gas-phase simulation by employing surrogate coefficients. In the model, non-electrolytic diOS was predicted and applied to prediction of LWC and $[H^+]$, which subsequently affect aerosol growth via aqueous reactions. Typically, the monoalkyl sulfate is identified as a product of the esterification of SA

5  with reactive species (Hettiyadura et al., 2015;Li et al., 2016a;Estillore et al., 2016;Chen et al., 2018). It is possible that monoalkyl sulfates can influence LWC and aerosol acidity differently from sulfuric acid, although they are strongly acidic and hygroscopic. Although Noziere et al. (2010) reported that OS could be produced by the reactions of GLY and sulfate radicals in the presence of aqueous AS under UV light, the amounts of formed monoalkyl OS and their influence on aerosol hygroscopicity is still not clear.

Some other factors in recent investigations, such as organic vapor wall loss and aerosol viscosity, have not accounted for by the UNIPAR model. The loss of organic vapor to the Teflon chamber wall can compete with the gas-particle partitioning process and the reactions in both the gas phase and aerosol phase to initiate a negative bias in the experimental measurements (Zhang et al., 2014;Mcvay et al., 2014). The modeling of the gas-wall process of semivolatile organic compounds can improve

the prediction of SOA mass in regional scales.  In addition, an increased aerosol viscosity via aging could modify the diffusivity of the partitioned organic molecules (Abramson et al., 2013) and the reaction rate constant for oligomerization in the aerosol phase.

## 5 Conclusions and implications

Despite numerous studies in SOA characterization and formation mechanisms, substantial biases between the simulated and

field-measured SOA mass were still found (Hodzic et al., 2016) due to the inadequacy of handling the dynamic multigenerational aging (Jathar et al., 2016) and aqueous reactions of the oxygenated products in the presence of an aqueous phase containing electrolytes (Ervens et al., 2011). In this study, the UNIPAR model addressed those issues using a dynamic age-driven $\alpha_i$ set, multiphase partitioning of organic compounds, and in-particle chemistry. Although the utilization of the age-driven $\alpha_i$ set improves the time series prediction of SOA mass, as shown in Fig. 3, the photochemical evolution of the gas-

phase products via monoalkylbenzene oxidation (Fig. 2, Fig. S2, and Fig. S3) does not increase the SOA mass, as is commonly suggested. Overall, the effect of an aqueous phase containing electrolytes on SOA formation was more critical than that of the $NO_x$ level under our simulated conditions. By adding a wet inorganic seed to the non-seed SOA system, the mass-base SOA yields under high $NO_x$ levels increase more than those under low $NO_x$ conditions (Fig. 6 in Section 4.4). The vapor pressure of volatile organonitrate and PAN-like species, which are formed at high $NO_x$ levels, are not low enough to increase partitioning

SOA mass (Fig. 7 A-II). Thus, SOA yields decreased by increasing $NO_x$ levels. Overall, both simulation and chamber data show that monoalkylbenzene SOA yields increase with a decreased alkyl chain length: toluene > ethylbenzene > n-



propylbenzene. This difference is most noticeable in the presence of an inorganic seed at high $NO_x$ levels (Fig. 7 A-II and H-II).

Due to the pervasiveness and relatively high concentration of toluene in the urban situation, where $HC/NO_x < 5.5$ and wet
5   inorganic seeds typically exist, the importance of toluene SOAs to the urban SOA burden can increase. The oxidation products from aromatic HCs can also involve cloud condensation nuclei activity due to their high reactivity via heterogeneous chemistry (Molteni et al., 2018), resulting in a change in the properties of clouds and fog and the urban radiation balance (Gordon et al., 2016). The unified aerosol-phase reaction rate constants for three monoalkylbenzenes represent the feasibility of applying the UNIPAR model to more aromatic systems (dialkyl benzenes and trialkyl benzenes) and the complex urban mixture.

*Acknowledgments.* This research was supported by the National Strategic Project-Fine particle of the National Research Foundation of Korea (NRF) funded by the Ministry of Science and ICT(MSIT), the Ministry of Environment (ME), and the Ministry of Health and Welfare (MOHW) (2017M3D8A1090654).



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



**Table 1: Experimental conditions and resulting SOA chamber data of the monoalkylbenzenes photooxidation experiments performed under various NO$_x$ conditions with or without inorganic seeded aerosol in the dual outdoor UF APHOR chambers.**

| Exp. ID [a] | Date [b] | HC (ppb) | NO$_x$ (HONO) (ppb) | Seeded aerosol [d] ($\mu g/m^3$) | HC/NO$_x$ (ppbC/ppb) | Y$_{SOA}$ [e] (%) | RH (%) | Temperature (K) | Note [f] |
|---|---|---|---|---|---|---|---|---|---|
| | | | | Initial condition | | | | | |
| Tol1 | 01/06/12 E [c] | 190 | 110 (40) | 50 | 12.1 | 18.9 | 18-81 | 280–306 | Fig. 6(a) |
| Tol2 | 01/06/12 W [c] | 190 | 95 (35) | - | 14.8 | 13.3 | 18-81 | 280–306 | Fig. 6(a) |
| Tol3 | 02/09/12 E [c] | 175 | 245 (35) | 46 | 5.0 | 15.3 | 21-83 | 280–307 | Fig. 6(d) |
| Tol4 | 02/09/12 W [c] | 180 | 246 (35) | - | 4.5 | 9.3 | 21-84 | 280–307 | Fig. 6(d) |
| Tol5 | 06/20/12 E [c] | 165 | 110 (15) | 35 (SA) | 10.5 | 15.6 | 27-83 | 295–317 | Fig. S7(a) |
| Tol6 | 12/16/17 E | 198 | 132 (79) | - | 10.5 | 8.6 | 23-58 | 283-300 | Fig. 3(a) and Fig. S7(b) |
| Tol7 | 02/25/18 W | 154 | 170 (22) | - | 6.4 | 3.3 | 20-44 | 293-313 | Fig. 3(b), Fig. S6(a) and Fig. S7(c) |
| Tol8 | 04/30/18 E | 127 | 306 (47) | 70 (SA) | 2.9 | 13.1 | 14-57 | 289-317 | Fig. 3(c), Fig. 5(a) and Fig. S7(d) |
| Tol9 | 06/14/18 W | 135 | 361 (80) | 130 (wAS) | 2.6 | 19.0 | 51-98 | 295-319 | Fig. S7(e) |
| EB1 | 12/05/17 E | 126 | 71 (32) | 43 | 14.2 | 15.4 | 18-57 | 287-310 | Fig. 6(b) and Fig. S6(b) |
| EB2 | 12/05/17 W | 134 | 74 (38) | - | 14.4 | 12.2 | 25-66 | 288-310 | Fig. 6(b) |
| EB3 | 01/04/18 E | 132 | 175 (13) | 50 | 6.0 | 21.8 | 30-85 | 267-291 | Fig. S8(a) |
| EB4 | 01/04/18 W | 131 | 175 (22) | - | 6.0 | 12.8 | 48-93 | 267-289 | Fig. S8(a) |
| EB5 | 12/10/17 E | 131 | 363 (13) | 39 | 2.9 | 10.1 | 20-83 | 271-298 | Fig. 6(e) |
| EB6 | 12/10/17 W | 128 | 363 (15) | - | 2.8 | 4.1 | 33-86 | 272-295 | Fig. 6(e) and Fig. 6(b) |
| EB7 | 02/19/18 W | 125 | 81 (36) | 80 (SA) | 12.3 | 25.6 | 19-46 | 292-315 | Fig. 5(b) and Fig. S8(b) |
| EB8 | 02/19/18 E | 112 | 63 (36) | 35 (dAS) | 14.3 | 11.0 | 13-39 | 292-314 | Fig. S8(b) |
| EB9 | 01/19/18 W | 169 | 106 (30) | 40 (wAS) | 12.7 | 28.6 | 20-87 | 269-302 | Fig. S8(c) |
| PB1 | 03/04/18 E | 100 | 87 (19) | 57 | 10.4 | 7.4 | 11-54 | 279-306 | Fig. 6(c) and Fig. S6(c) |
| PB2 | 03/04/18 W | 109 | 108 (24) | - | 9.1 | 5.4 | 17-59 | 279-305 | Fig. 6(c) |
| PB3 | 03/28/18 E | 87 | 264 (36) | 54 | 3.0 | 7.1 | 11-43 | 285-312 | Fig. 6(f) |
| PB4 | 03/28/18 W | 88 | 248 (33) | - | 3.2 | 4.6 | 16-51 | 285-312 | Fig. 6(f) |
| PB5 | 04/05/18 W | 101 | 76 (35) | 70 (SA) | 12.0 | 15.7 | 30-93 | 282-312 | Fig. 5(c) and Fig. S9(a) |
| PB6 | 04/17/18 E | 101 | 149 (141) | 70 (SA) | 6.1 | 11.9 | 14-85 | 278-313 | Fig. S9(b) |
| PB7 | 04/17/18 W | 101 | 155 (126) | 70 (wAS) | 5.9 | 18.1 | 40-91 | 279-310 | Fig. S9(b) |
| PB8 | 06/14/18 E | 83 | 353 (148) | 90 (SA) | 2.1 | 10.7 | 22-90 | 294-322 | Fig. S9(c) |

[a] "Tol", "EB", and "PB" represent toluene, ethylbenzene, and n-propylbenzene oxidation experiments, respectively.

5    [b] "E" or "W" denotes the east or west chamber.

[c] SOA data obtained from Im et al. (2014).

[d] "SA", "wAS", and "dAS" denote sulfuric acid seeds, wet ammonium sulfate seeds, and dry ammonium sulfate seeds. (dry: RH < ERH; wet: RH > ERH). For those without notification, SO$_2$ (in the unit of ppb) was injected into the chamber to generate acidic seeds.

10    [e] SOA yield is estimated using Y$_{SOA}$ = ΔOM/ΔHC, where ΔOM is formed organic matter, ΔHC is consumed HC. Yield in the table was estimated where SOA mass reached to the maximum over the course of the experiments.

[f] This column denotes that the corresponding data was used in which figures.

The accuracy of RH is 5 %. The accuracy of temperature is 0.5 K.



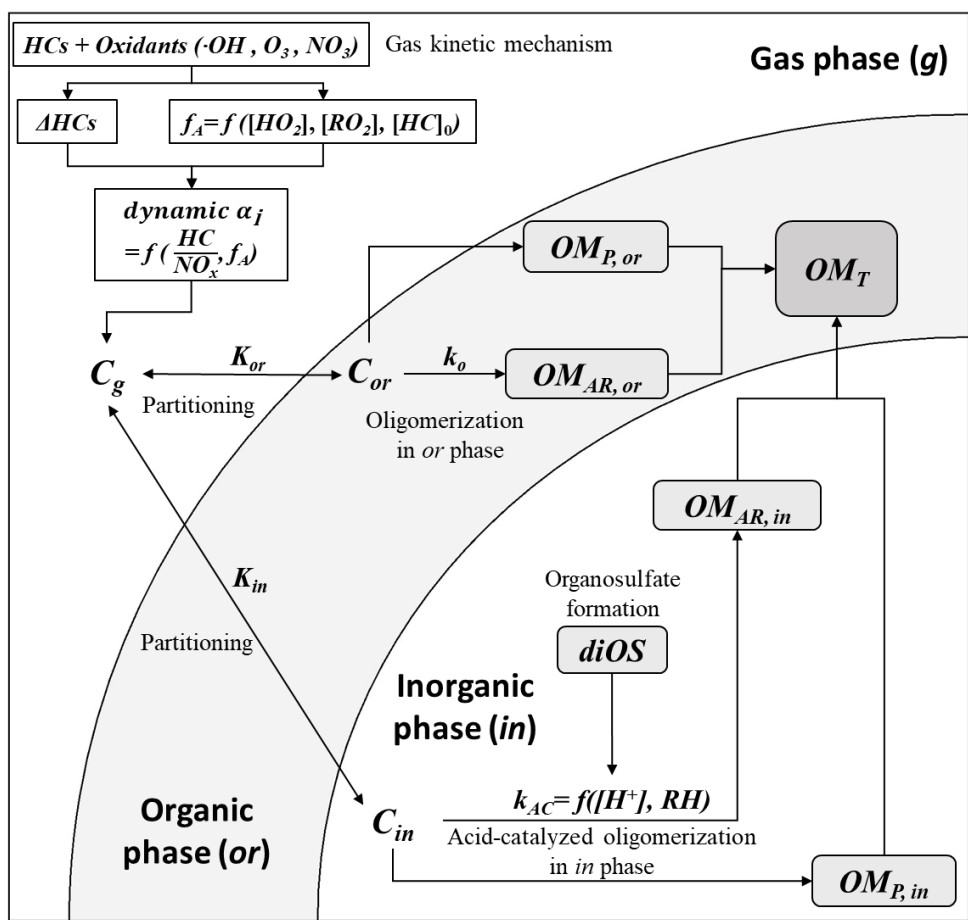

**Figure 1: Simplified scheme of the UNIPAR model.** $[HC]_0$ represent the initial hydrocarbon (HC) concentration. The dynamic mass-based stoichiometric coefficient (dynamic $\alpha_i$), the consumption of HCs ($\Delta$HCs), and the concentrations of hydroperoxide radical ($[HO_2]$) and organic peroxyl radical ($[RO_2]$) are simulated from the gas kinetic model. The aging factor ($f_A$) is represented as a function of $[HO_2]$, $[RO_2]$, and $[HC]_0$, which is detailed in Section 3.1. $C$ and $K$ denote the concentration and the partitioning coefficient in gas phase ($g$), organic phase ($or$), and inorganic phase ($in$). $k_{or,i}$ denotes the reaction rate constant of oligomerization in $or$ phase. $k_{AC,i}$ denotes the reaction rate constant of acid-catalyzed oligomerization in $in$ phase and is represented as a function of aerosol acidity ($[H^+]$) and ambient humidity (RH). $OM$ represents the concentration of organic matter. Subscripts, "$AR$", "$P$", and "$T$" denote aerosol-phase reactions, partitioning, and total organic matter, respectively. Subscript $i$ represents each lumping species. $diOS$ represents the concentration of organosulfate (dialkyl sulfate ($diOS$) in this study).



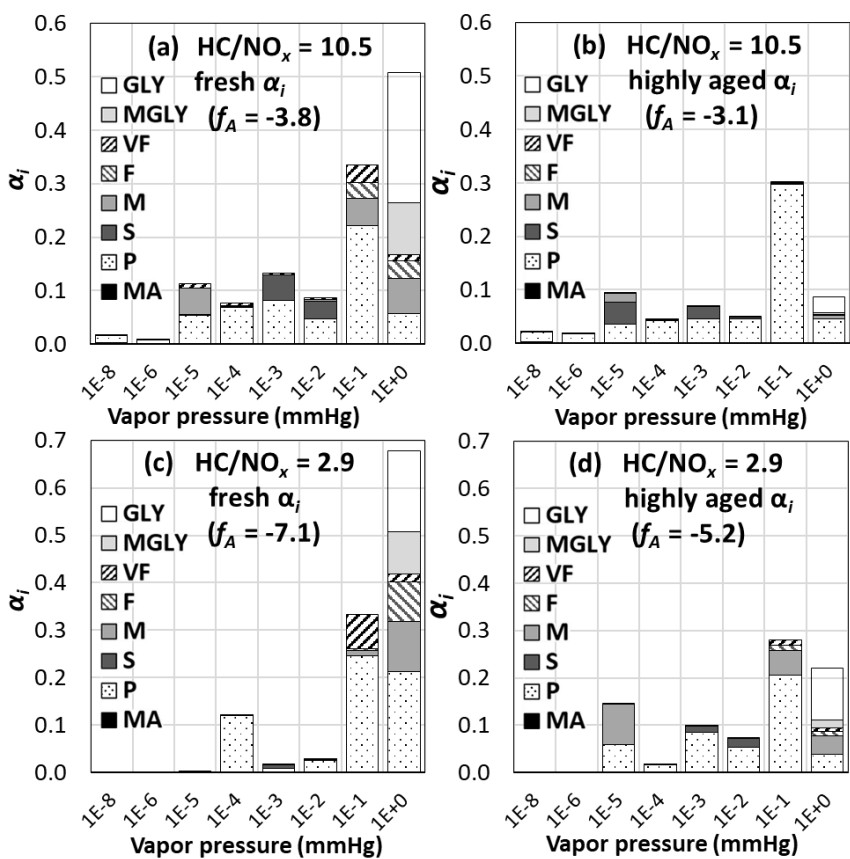

**Figure 2:** The mass-based stoichiometric coefficients ($\alpha_i$) of each species, $i$, from toluene oxidation under low $NO_x$ level (simulation based on the sunlight of Exp. Tol6, HC/$NO_x$ = 10.5, 12/16/18) at (a) fresh condition and (b) highly aged condition, and under high $NO_x$ level (simulation based on the sunlight of Exp. Tol8, HC/$NO_x$ = 2.9, 04/30/18) at (c) fresh condition and (d) highly aged condition, where $f_A$ is the aging factor as derived in Eq. 1. The oxygenated products predicted by the explicit gas kinetic model are lumped as a function of vapor pressure (8 groups: $10^{-8}$, $10^{-6}$, $10^{-5}$, $10^{-4}$, $10^{-3}$, $10^{-2}$, $10^{-1}$, and 1 mmHg) and aerosol phase reactivity (6 groups), i.e., very fast (VF: tricarbonyls and $\alpha$-hydroxybicarbonyls), fast (F: 2 epoxides or aldehydes,), medium (M: 1 epoxide or aldehyde), slow (S: ketones), partitioning only (P), and multialcohol (MA). MGLY (methylglyoxal) and GLY (glyoxal) were lumped separately due to the relatively high reactivity.





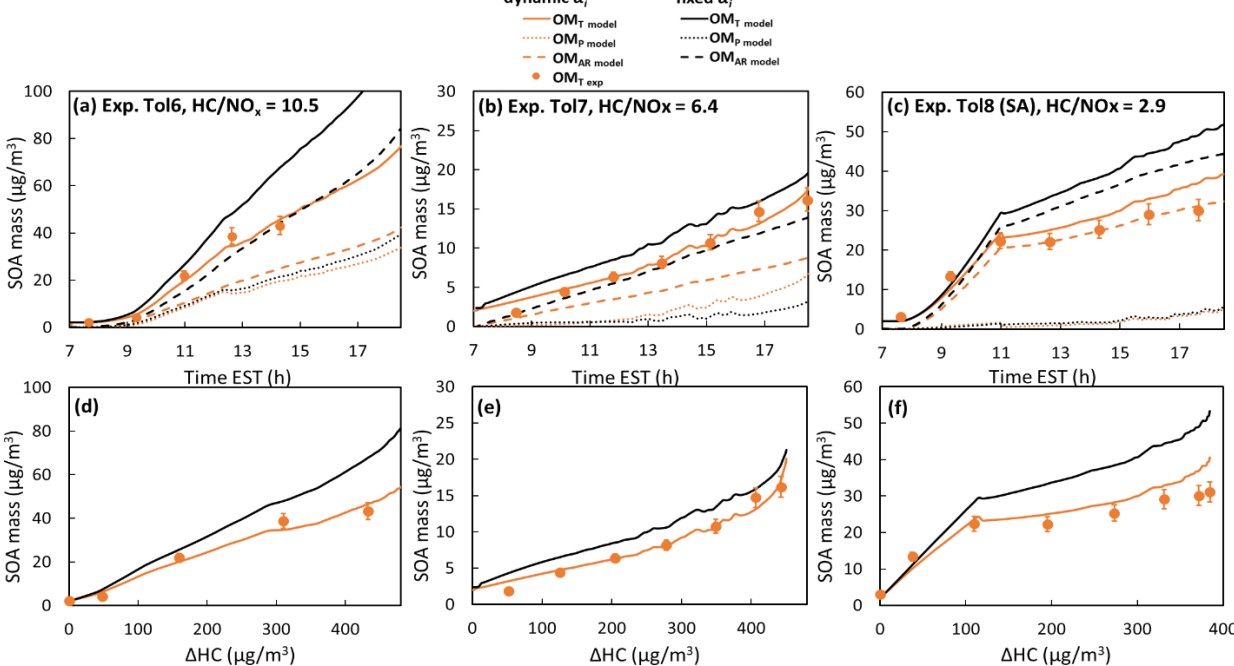

**Figure 3: Comparison between simulated SOA mass by the fixed $\alpha_i$ ($\alpha$ fixed at the point of HC being consumed half of the total consumption) and dynamic $\alpha_i$ ($\alpha$ evolving as photooxidation) under (a) low $NO_x$ condition (Exp. Tol6, HC/$NO_x$ = 10.5), (b) moderate $NO_x$ condition (Exp. Tol7, HC/$NO_x$ = 6.4), and (c) high $NO_x$ condition (Exp. Tol8, HC/$NO_x$ = 2.9 with sulfuric acid (SA) seeded aerosol). (d), (e), and (f) represent the time-dependent SOA growth curve (SOA mass concentration against consumed HC) corresponding to the experimental conditions of (a), (b), and (c), respectively. The solid circle represents the experimental measurements. The SOA mass is corrected for particle loss to the chamber wall. The experimental conditions are available in Table 1.**



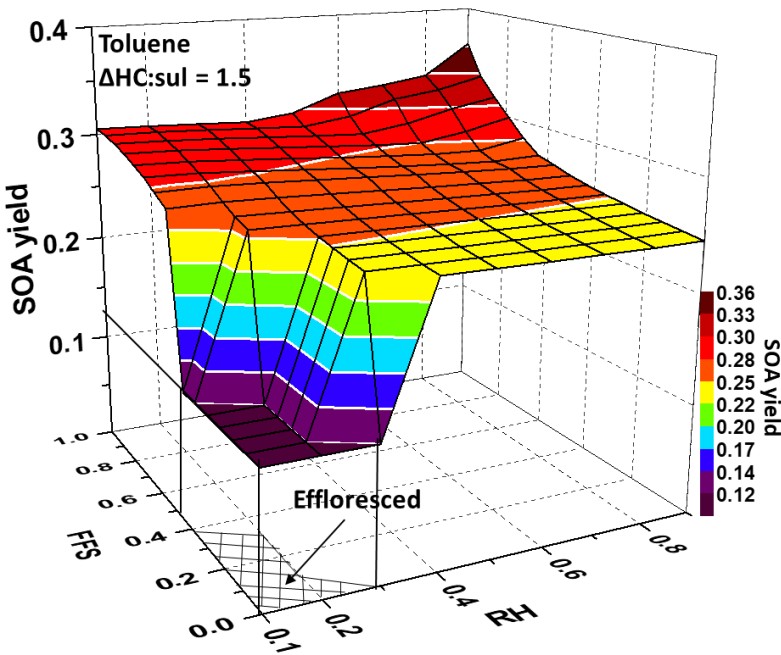

**Figure 4: Simulated toluene SOA yields ($Y_{SOA}$ = ΔOM/ΔHC) as a function of relative humidity (RH: 0.1 ~ 0.9) and fractional free sulfate (FFS: 0 ~ 1), where FFS = ([SO$_4^{2-}$]-0.5[NH$_4^+$])/[SO$_4^{2-}$] and is used to estimate aerosol acidity ([H$^+$]) in inorganic thermodynamic model. The RH and FFS are fixed in the simulations. The gas-phase simulations are based on the experimental condition of 06/14/2018 (Exp. Tol9 in Table 1) (initial HC concentration = 20 ppb, HC/NO$_x$ = 2, sulfate mass concentration = 20 μg/m$^3$, and the mass ratio of the consumed HC to sulfate (ΔHC:sulf) = 1.5).**





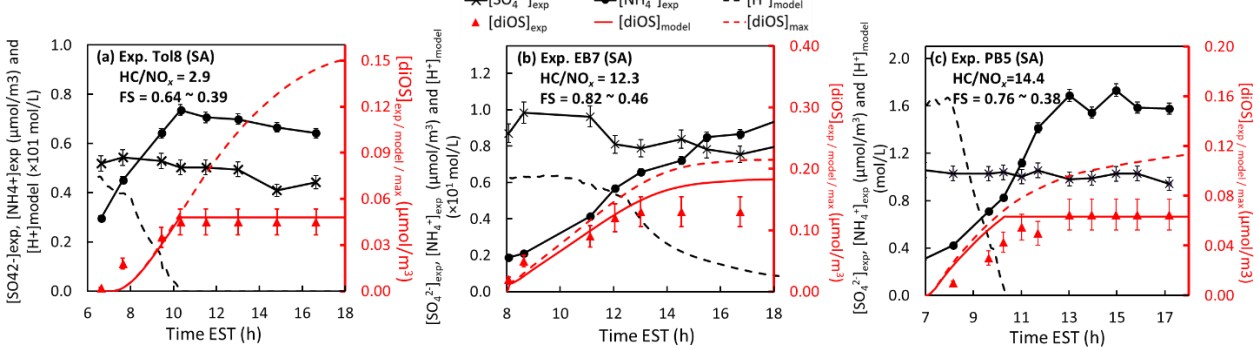

**Figure 5: Time profiles of measured inorganic sulfate concentration ([SO$_4^{2-}$]$_{exp}$), ammonium concentration ([NH$_4^+$]$_{exp}$), diOS concentration ([diOS]$_{exp}$), the predicted proton concentration ([H$^+$]), diOS concentration ([diOS]$_{model}$), and the maximum diOS concentration ([diOS]$_{max}$) (assuming there is no ammonia neutralization in the system) for SOA generated from (a) toluene (HC/NO$_x$ = 2.9, OM-to-sulfate mass ratio (OM:sulf) = 1.4), (b) ethylbenzene (HC/NO$_x$ = 12.3, OM:sulf = 1.4), and (c) n-propylbenzene (HC/NO$_x$ = 14.4, OM:sulf = 0.7). The degree of neutralization is indicated by FS, which is ranging from 1 (for sulfuric acid) to 0.33 (for ammonium sulfate). "SA" stands for experiment with direct-injection sulfuric acid seeded aerosols. The ion and diOS concentrations were corrected for the particle loss to the chamber wall. The experimental conditions are available in Table 1.**





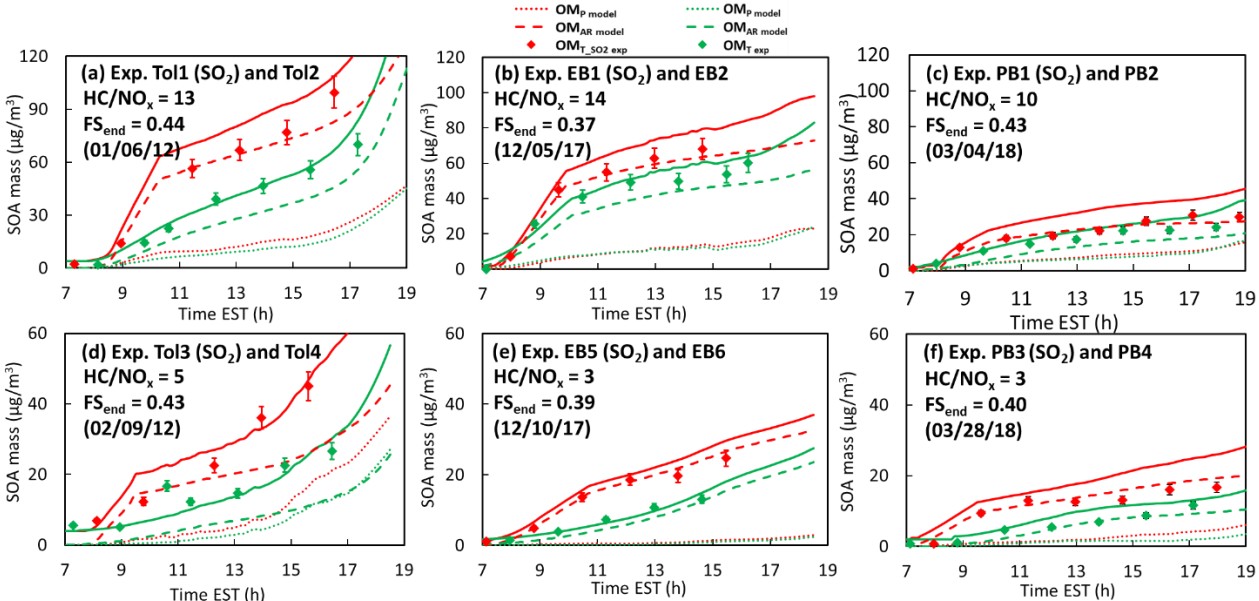

**Figure 6: Time profiles of measured and modeled SOA mass concentrations for toluene, ethylbenzene, and n-propylbenzene SOA under low/high $NO_x$ conditions in the presence/absence of $SO_2$-derived sulfuric acid seeded aerosol. The red and green colors indicate experiments with and without $SO_2$, respectively. Solid, dashed, and dotted lines denote the total organic matter ($OM_T$), the OM from partitioning only ($OM_P$), and the OM from the aerosol-phase reactions ($OM_{AR}$), respectively. The degree of ammonia neutralization with sulfuric acid is indicated by the $FS_{end}$, which is the FS at the end of the experimental run. The $FS_{end}$ is ranging from 1 (for sulfuric acid) to 0.33 (for ammonium sulfate). The uncertainty associated with experimentally measured OM is about 9 %. The SOA mass was corrected for the particle loss to the wall. The experimental conditions are available in Table 1.**

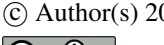


| Conditions \ ID | A | B | C | D | E | F | G | H | I |
|---|---|---|---|---|---|---|---|---|---|
| RH (%) | 45 | 65 | 45 | 65 | 45 | 45 | 45 | 45 | 45 |
| Temperature (K) | 298 | 298 | 298 | 298 | 273 | 298 | 273 | 298 | 298 |
| Seed condition (μg/m³) | NH₄HSO₄ (20) | NH₄HSO₄ (20) | (NH₄)₂SO₄ (20) | (NH₄)₂SO₄ (20) | NH₄HSO₄ (20) | No seed | No seed | NH₄HSO₄ (20) | No seed |
| HC/NOₓ (ppbC/ppb) | 2 | 2 | 2 | 2 | 2 | 2 | 2 | 10 | 10 |
| | High NOₓ condition | | | | | | | Low NOₓ condition | |



**Figure 7:** The simulated SOA mass (Panel I) for toluene (Tol), ethylbenzene (EB) and n-propylbenzene (PB) under different conditions, as are listed the top table. The initial concentrations of monoalkylbenzenes, pre-existing OM (OM₀), NH₄HSO₄ (AHS) seeded aerosol, and (NH₄)₂SO₄ (AS) seeded aerosol are set at 20 ppb, 2 μg/m³, 20 μg/m³, and 20 μg/m³, respectively. The gas-phase simulation used the sunlight on 06/14/2018 (Exp. Tol9 in Table 1). OMₚ and OMₐᵣ represent the organic matter from the partitioning process and aerosol-phase reactions. (Panel II) shows the time-dependent SOA growth curve for three monoalkylbenzenes under corresponding conditions (top table). The concentrations in the legends refer to the mass concentrations of the consumed hydrocarbons in each simulation under the high/low NOₓ conditions.