# Peer review of "Simulation of SOA Formation from the Photooxidation of Monoalkylbenzenes in the Presence of Aqueous Aerosols Containing Electrolytes under Various NOx Levels"

_Atmospheric Chemistry and Physics, 2018_

## Referee Comment (RC1) · K. Gorkowski (Referee) · 22 Nov 2018

**Review of:**

*Simulation of SOA Formation from the Photooxidation of Monoalkylbenzenes in the Presence of Aqueous Aerosols Containing Electrolytes under Various NOx Levels.*

[Figure]

**1  General Comments**

The manuscript by Zhou, Jang, and Yu present an improvement to the Unified Partitioning-Aerosol Phase Reaction (UNIPAR) model. The authors added an age-driven mass-based stoichiometric coefficient $(\alpha_i)$, to predict the temporal evolution of the aerosol system. They compare the new UNIPAR model to experiments conducted in the UF APHOR chambers.

The model improvement and experiments focused on aromatic molecules, specifically toluene, ethylbenzene, and n-propylbenzene. They used both seeded and non-seeded ammonium sulfate experiments as well as, wet vs. dry seeds. The updated UNIPAR model showed remarkably good agreement when a dynamic $\alpha_i$ was included in the model.

The manuscript is suited for Atmospheric Chemistry and Physics and is of interest to the community. It is well written and highlights essential insights into the partitioning and chemical reactions in multiphase aerosol particles. I have a minor comment regarding the dynamic $\alpha_i$, and a few line comments. With these minor concerns addressed, I would recommend this manuscript for publication.

**2  Specific: Dynamic $\alpha_i$**

I understand the mass-based stoichiometric coefficient $(\alpha_i)$, has to be dynamic to capture the full evolution of the aerosol mass. It is not clear on page 5 line 7, if the dynamic reconstruction is a fit to smog chamber data or not. Section 3.1 reads as if $\alpha_i$ was fitted at the beginning and ending conditions of the experiment. Then assuming that is correct, does $\alpha_i$ have any value other than a free parameter?

Since $\alpha_i$ was the major factor that brought the experiments and model into an agreement, is this fit general for the atmosphere or system specific?

**3 Line Comments**

- Page 1 line 13: "applied to estimate" would be clearer if changed to "used to estimate"

- Page 1 line 19: Shouldn't the importance of electrolytes over NOx or $\alpha_i$, be expected or is this new insight?

- Page 1 line 21 and Page 11 line 14: "presence of wet electrolytic seeds" is this mainly the salting-in effect (and not chemical reactions) that causes the increase in SOA mass? From, Figure 7 the small fraction of $OM_{AR}$ in A-D seems to suggest that is the case. Have you ran simulations at higher RHs, say 90%?

- Page 4 line 10: There are theoretical calculations to include in the support the assumption of phase separation. See Zuend, A. and Seinfeld, J. H.: Modeling the gas-particle partitioning of secondary organic aerosol: The importance of liquid-liquid phase separation, Atmos. Chem. Phys., 12(9), 3857–3882, doi:10.5194/acp-12-3857-2012, 2012.

- Page 5 line 26: How did you settle on this formula for the activity coefficients? I suggest adding that discussion to the SI.

- Page 11 Line 3: "RH is insignificant" only at these experimental conditions. Maybe change to "RH is insignificant for our experiments, discussed in Section 4.2."

- Figure 5: I find the figure's y-axis labels a bit cramped. Add a little more white space between the three panels to improve readability.

---

## Referee Comment (RC2) · Anonymous Referee #2 · 9 Feb 2019

General Comments

This manuscript describes high-quality modeling efforts simulating a series of sunlit chamber experiments on toluene, ethylbenzene, and propylbenzene oxidation. The chamber experiments were conducted at varying RH levels so that the inorganic phase of aerosol would be liquid or solid at different points. NOx levels were also varied from experiment to experiment. The UNIPAR model was extended by the introduction of 6 reactivity bins within each volatility bin, along with separate treatment for glyoxal, methylglyoxal, and isoprene epoxidiol products. Futhermore, the mass-based stoichio-
metric coefficients of products were allowed to evolve between coefficients measured on fresh aerosol to those measured on aged aerosol. With these extensions, the model was able to simulate chamber data closely, while allowing the authors to compare the relative importance of environmental factors on SOA yields. They conclude that for monoalkylbenzene oxidation systems, the presence of an aqueous aerosol phase has a larger positive effect on SOA yields that switching from high NOx to low NOx conditions. This work will be of interest to a broad audience of scientists with interests in particulate air pollution and its mitigation.

Specific Comments

It is unclear which experiments were conducted in sunlight, and how this important factor affected the results. The paper mentions that the sunlight from one experiment performed near summer solstice was used in gas-phase simulations, apparently for all experiments, even those conducted in mid-winter. Were all experiments performed on clear days? How might using more intense sunlight in simulations of winter experiments affect the uncertainties of the results?

p. 6 line 10: Was oligomerization in both the organic phase and the aqueous (inorganic) phase based on self-dimerization of individual products, or lumped products? In other words, were cross-reactions possible between molecules lumped into a single bin?

p. 10 line 32: The authors identify a large temperature effect on SOA yields, as seen in other studies. However, the authors are uniquely positioned to identify whether this effect is due only to partitioning, as typically assumed, or is also due to temperature-dependent reactions that either destroy condensable species or produce other species that hinder gas-to-particle transfers. Partitioning seems to be such a minor SOA source in this study that it is surprising that the observed temperature effects are so pronounced. Could the authors probe the cause of the temperature effect?

In Figure 5, dialkyl organosulfates (diOS) concentrations seem to track ammonium concentrations. In one experiment, the authors comment that diOS formation ceases

[Figure]

when the aerosol effloresces. Is there any causative relationship between diOS and ammonium concentrations in wet aerosol?

Figure 7 implies that in most experiments, aqueous-phase SOA production is much greater than SOA production via traditional partitioning mechanisms, but the authors don't seem to make this comparison or comment on it. Is it a fair comparison?

Figure S1 indicates that ozone is generated too quickly in the model, sometimes by a factor of 2 or 3. Can the authors comment on the implications of this overprediction? Is it related to the "artificial OH radicals" added to the model in certain experiments?

Technical Corrections

P. 2 line 5: Unclear comparison: " . . . fewer global emissions" that what?

p. 2 line 11: When the authors refer to "regional weather" do they really mean climate change?

p. 4 line 9: unclear comparison with literature: What system was "the reported value of 2" measured for? Is it a toluene oxidation study?

p. 6 line 18: another unclear comparison with literature: What system was "the reported value of 1.4 g/cm3" measured for?

p. 7 line 2: It appears that "irreversibility and nonvolatility" should be "irreversibly formed and nonvolatile".

p. 9 line 12: The meaning of the phrase "contributed to" is unclear in this sentence.

---

## Author Comment (AC1) · 14 Mar 2019

Chufan Zhou, Myoseon Jang, and Zechen Yu

mjang@ufl.edu

We thank reviewer Dr. Gorkowski for the valuable comments on the manuscript.

**Major comment: Dynamic $\alpha_i$**

I understand the mass-based stoichiometric coefficient ($\alpha_i$), has to be dynamic to capture the full evolution of the aerosol mass. It is not clear on page 5 line 7, if the dynamic reconstruction is a fit to smog chamber data or not. Section 3.1 reads as if $\alpha_i$ was fitted at the beginning and ending conditions of the experiment. Then assuming that is correct, does $\alpha_i$ have any value other than a free parameter? $\alpha_i$ was the major factor that brought the experiments and model into an agreement, is this fit general for the atmosphere or system specific?

**Response:**

The dynamic reconstruction of $\alpha_i$ is not produced by fitting SOA simulation with $\alpha_i$ to the smog chamber data. The dynamic $\alpha_i$ is created by compositing the two $\alpha_i$ sets at the fresh gas composition (fresh $\alpha_i$) and the highly aged gas composition (highly aged $\alpha_i$). The fresh $\alpha_i$ and the highly aged $\alpha_i$ are predicted using the predetermined equations, which are a function of NO$_x$ level near the summer solstice (June 14[th], 2018). At a given NO$_x$ level, dynamic $\alpha_i$ is reconstructed using the aging scale factors ($f_A(t) = log \frac{[HO_2]+[RO_2]}{[HC]_0}$). Under a given NO$_x$ level (HC ppbC/NO$_x$ ppb) the $f_A(t)$ is maximized late afternoon (~4 PM) at near solstice. For the fresh condition, $f_A(fresh)$ is determined at 20% of total hydrocarbon consumption. $f_A(fresh)$ and $f_A(higly\ aged)$ are $f_A(t)$ values at fresh and highly aged conditions, respectively. For example, $f_A(fresh)$ and $f_A(higly\ aged)$ at HC/NO$_x$ = 45 for toluene are -3.7 and -2.9, respectively. At HC/NO$_x$ = 2, $f_A(fresh)$ and $f_A(higly\ aged)$ are -7.2 and -5.2, respectively. We define aging factor ($f_A'(t)$) at time = t as follows

$$f_A'(t) = \frac{f_A(highly\ aged) - f_A(t)}{f_A(highly\ aged) - f_A(fresh)}$$

Then, the $\alpha_i$ set is dynamically reconstructed by a weighted average method (Eq. (2) in the manuscript) using fresh $\alpha_i$ set, highly aged $\alpha_i$ set. and $f_A'(t)$.

**Minor Comments:**

**(1) Page 1 line 13:** "applied to estimate" would be clearer if changed to "used to estimate"

**Response:**

This has been corrected in the revised manuscript.

**(2) Page 1 line 19:** Shouldn't the importance of electrolytes over $NO_x$ or $\alpha_i$, be expected or is this new insight?

**Response:**

The impact of hygroscopic seed, $NO_x$, or aging on SOA growth has been studied by numerous researchers. However, the relative importance of these variables on SOA was not well investigated. Based on our chamber studies and simulation results (Figure 7), we conclude that the effect of an aqueous phase containing electrolytes on SOA yields was more important than that of the $NO_x$ level under our simulated conditions or the utilization of the age-driven $\alpha_i$ set.

**(3) Page 1 line 21 and Page 11 line 14:** "presence of wet electrolytic seeds" is this mainly the salting-in effect (and not chemical reactions) that causes the increase in SOA mass? From, Figure 7 the small fraction of $OM_{AR}$ in A-D seems to suggest that is the case. Have you run simulations at higher RHs, say 90%?

**Response:**

Although some compounds (e.g., glyoxal) can be salting in (Kampf et al., 2013). In general, electrolytic inorganic salts results in salting out for most organic compounds (Wang et al., 2014). In this paper, the organic solubility in the salted aqueous phase was predicted using the predetermined polynomial equation, which was produced using the solubility (activity coefficient) of a variety of model compounds, which were parameterized with molecular weight (MW) and organic to carbon ratio (O:C) at different humidity and inorganic compositions. Evidently, the activity coefficient of most organic compound increases as increasing salt concentrations (decreasing humidity) supporting a salting out effect (Section S3 in supporting information). The

sign of the coefficient for humidity in equation 4 is negative. In the revision of the manuscript, equation 4 was updated by including more model compounds and reads now,

$$\gamma_{in,i} = e^{4.789 \cdot ln(MW_i) - 4.701 \cdot ln(O:C_i) - 5.484 \cdot FS - 0.098 \cdot (100 \cdot RH) - 12.464}$$

**(4) Page 4 line 10:** There are theoretical calculations to include in the support the assumption of phase separation. See Zuend, A. and Seinfeld, J. H.: Modeling the gas-particle partitioning of secondary organic aerosol: The importance of liquid-liquid phase separation, Atmos. Chem. Phys., 12(9), 3857–3882, doi:10.5194/acp-12-3857-2012, 2012.

**Response:**

We cited the original paper in the manuscript at page 4 line 11.

**(5) Page 5 line 26:** How did you settle on this formula for the activity coefficients? I suggest adding that discussion to the SI.

**Response:**

In order to provide better description, Section S3 ("Activity coefficient of organic species in the aqueous phase containing electrolytes") has been revised and reads now,

"In the UNIAPR model, the formation of aromatic SOA is simulated with the assumption of organic-inorganic phase separation. To predict the partitioning of organic species on both the organic phase and the inorganic phase, the key model parameters are $K_{or,i}$ and $K_{in,i}$, respectively (described in Section 3.2 of the main manuscript). In order to predict $K_{in,i}$, the calculation of the activity coefficient ($\gamma_{in,i}$) of organic species in the inorganic phase (aqueous phase containing electrolytes) is necessary.

In our study, $\gamma_{in,i}$ was semi-empirically predicted by a polynomial equation, which was fit the theoretical $\gamma_{in,i}$ of various organic compounds to relative humidity ($RH$), fractional sulfate ($FS$), and molecular structures (i.e., molecular sizes ($MW_i$) and polarity ($O:C_i$)). The theoretical $\gamma_{in,i}$ was determined at the maximum solubility of organic species in the electrolytic aqueous phase ($SO_4^{2-}$-$NH_4^+$-$H_2O$ system) using the Aerosol Inorganic-Organic Mixtures Functional Groups Activity Coefficients (AIOMFAC) (Zuend et al., 2011). AIOMFAC was run for the estimation of $\gamma_{in,i}$ of 26 model compounds with diverse $MW_i$ and $O:C_i$ under varying inorganic phase compositions (FS and hygroscopicity linked to RH). The oligomeric products form in aqueous phase, but they deposit to the organic phase due to their poor solubility in inorganic phase.

However, some hydrophilic oligomers can dissolve in both organic and inorganic phases. For example, glyoxal-origin oligomers might be hydrophilic and partially soluble in inorganic phase. Hence, the trace amount of glyoxal-oligomer (MW = 290 g/mol and O:C = 1 with mole fraction = 0.01) was included in inorganic phase as seen in Table S4. In Figure S1, the $\gamma_{in,i}$ predicted by AIOMFAC was plotted to that predicted by the polynomial equation (Eq. 4 in the manuscript) along with the one-to-one line for 26 organic species (Table S4). FS ranges from 0.34 to 1.0 and RH ranges from 0.1 to 0.8."

**Table S4: The molecular structures of the oligomeric compound (a) and 26 model compounds (b) with O:C ratios and MW, which were employed to derive the polynomial equation to predict $\gamma_{in,i}$ of organic species in electrolytic aqueous phase. The name of the organic compound with symbol * originates from MCM website (http://mcm.leeds.ac.uk/MCMv3.3.1/home.htt).**

[Figure]

**(a)**

| # | Oligomer |
|---|---|
| Structure | |
| # of GLY | n=5 |
| Formula | $C_{10}H_{10}O_{10}$ (n=5, nonhydrate on both end) |
| O:C | 1.000 |
| MW | 290 |

**(b)**

| # | 1 | 2 | 3 | 4 | 5 | 6 | 7 |
|---|---|---|---|---|---|---|---|
| Structure | | | | | | | |
| Name | phenethyl alcohol | o-cresol | 2-methoxy-2-methylpropane | MCATECHOL* | 1,7-heptanediol | phenylacetic acid | norpinic acid |
| Formula | C8H10O | C7H8O | C5H12O | C7H8O2 | C7H16O2 | C8H8O2 | C8H12O4 |
| O:C | 0.125 | 0.143 | 0.200 | 0.286 | 0.286 | 0.250 | 0.500 |
| MW | 122.167 | 108.140 | 88.150 | 124.139 | 132.203 | 136.150 | 172.180 |

| # | 8 | 9 | 10 | 11 | 12 | 13 | 14 |
|---|---|---|---|---|---|---|---|
| Structure | | | | | | | |
| Name | C6OTKETOOH* | EBZBPER2OH* | 2-phenylethylhydroperoxide | MMALNHYOOH* | octanoic acid | tert-butanol | 1,4-dihydroxy-2-butene |
| Formula | C6H8O6 | C8H12O4 | C8H10O2 | C5H6O6 | C8H16O2 | C4H13O | C4H8O2 |
| O:C | 1.000 | 0.500 | 0.250 | 1.200 | 0.250 | 0.250 | 0.500 |
| MW | 176.124 | 172.180 | 138.166 | 162.097 | 144.214 | 77.147 | 88.106 |

| # | 15 | 16 | 17 | 18 | 19 | 20 | 21 |
|---|---|---|---|---|---|---|---|
| Structure | | | | | | | |
| Name | hexanedioic acid | octanol | TLEMUCOOH* | 1-pentanol | syringic acid | methylglyoxal | 3-methyl-4-propyl-octane-2,6-diol |
| Formula | C6H10O4 | C8H18O | C7H10O6 | C5H12O | C9H10O5 | C3H4O2 | C12H26O2 |
| O:C | 0.667 | 0.125 | 0.857 | 0.200 | 0.556 | 0.667 | 0.167 |
| MW | 146.142 | 130.231 | 190.151 | 88.150 | 198.174 | 72.063 | 202.338 |

| # | 22 | 23 | 24 | 25 | 26 | | |
|---|---|---|---|---|---|---|---|
| Structure | | | | | | | |
| Name | 1-propanol | glyoxal | EBZBPEROOH* | EBENZOLOOH* | ferulic acid | | |
| Formula | C3H8O | C2H2O2 | C8H12O5 | C8H12O6 | C10H10O4 | | |
| O:C | 0.333 | 1.000 | 0.625 | 0.750 | 0.400 | | |
| MW | 60.096 | 58.000 | 188.179 | 204.178 | 194.186 | | |

[Figure]

**Figure S1: The $\gamma_{in,i}$ predicted by AIOMFAC was plotted to that predicted by the polynomial equation (Eq. 4 in the manuscript) along with the one to one line.**

**(6) Page 11 Line 3:** "RH is insignificant" only at these experimental conditions. Maybe change to "RH is insignificant for our experiments, discussed in Section 4.2."

**Response:**

We changed the sentence to "… the effect of RH on SOA growth is insignificant in our simulation, discussed in Section 4.2."

**(7) Figure 5:** I find the figure's y-axis labels a bit cramped. Add a little more white space between the three panels to improve readability.

**Response:**

Figure 5 has been revised based on the comment the reviewer as follows,

[Figure]

**Figure 5: Time profiles of measured inorganic sulfate concentration ([SO$_4^{2-}$]$_{exp}$), ammonium concentration ([NH$_4^+$]$_{exp}$), diOS concentration ([diOS]$_{exp}$), the predicted proton concentration ([H$^+$]), diOS concentration ([diOS]$_{model}$), and the maximum diOS concentration ([diOS]$_{max}$) (assuming there is no ammonia neutralization in the system) for SOA generated from (a) toluene (HC/NO$_x$ = 2.9, OM-to-sulfate mass ratio (OM:sulf) = 1.4), (b) ethylbenzene (HC/NO$_x$ = 12.3, OM:sulf = 1.4), and (c) n-propylbenzene (HC/NO$_x$ = 14.4, OM:sulf = 0.7). The degree of neutralization is indicated by FS, which is ranging from 1 (for sulfuric acid) to 0.33 (for ammonium sulfate). "SA" stands for experiment with direct-injection sulfuric acid seeded aerosols. The ion and diOS concentrations were corrected for the particle loss to the chamber wall. The experimental conditions are available in Table 1.**

**Reference**

Kampf, C. J., Waxman, E. M., Slowik, J. G., Dommen, J., Pfaffenberger, L., Praplan, A. P., Prévôt, A. S. H., Baltensperger, U., Hoffmann, T., and Volkamer, R.: Effective Henry's Law Partitioning and the Salting Constant of Glyoxal in Aerosols Containing Sulfate, Environmental Science & Technology, 47, 4236-4244, 10.1021/es400083d, 2013.

Wang, C., Lei, Y. D., Endo, S., and Wania, F.: Measuring and Modeling the Salting-out Effect in Ammonium Sulfate Solutions, Environmental Science & Technology, 48, 13238-13245, 10.1021/es5035602, 2014.

Zuend, A., Marcolli, C., Booth, A. M., Lienhard, D. M., Soonsin, V., Krieger, U. K., Topping, D. O., McFiggans, G., Peter, T., and Seinfeld, J. H.: New and extended parameterization of the thermodynamic model AIOMFAC: calculation of activity coefficients for organic-inorganic mixtures containing carboxyl, hydroxyl, carbonyl, ether, ester, alkenyl, alkyl, and aromatic functional groups, Atmospheric Chemistry and Physics, 11, 9155-9206, 10.5194/acp-11-9155-2011, 2011.

---

## Author Comment (AC2) · 14 Mar 2019

Chufan Zhou, Myoseon Jang, and Zechen Yu

mjang@ufl.edu

We appreciate the reviewer for the valuable comments.

**Major comments:**

**(1)** It is unclear which experiments were conducted in sunlight, and how this important factor affected the results. The paper mentions that the sunlight from one experiment performed near summer solstice was used in gas-phase simulations, apparently for all experiments, even those conducted in mid-winter. Were all experiments performed on clear days? How might using more intense sunlight in simulations of winter experiments affect the uncertainties of the results?

**Response:**

Please also find the response to the major comment from reviewer 1. All the simulations were performed by using natural sunlight. As described in Section 3.1 in the manuscript, the aging scale factor is defined as $f_A(t) = log\frac{[HO_2]+[RO_2]}{[HC]_0}$. In order to set the aging scale factor for the fresh ($f_A(fresh)$) and highly aged processing ($f_A(higly\ aged)$) of hydrocarbon oxidation, we utilized the sunlight intensity near summer solstice (on 06/14/2018 with a clear sky in Gainesville, Florida (29.64185° N, 82.347883° W)). The aging factor ($f_A'(t)$) at time = t is estimated by using the equation below,

$$f_A'(t) = \frac{f_A(highly\ aged) - f_A(t)}{f_A(highly\ aged) - f_A(fresh)}$$

Then, the $\alpha_i$ set is dynamically reconstructed by a weighted average method (Eq. (2) in the manuscript) using fresh $\alpha_i$ set, highly aged $\alpha_i$ set and $f_A'$(t).

**(2)** p. 6 line 10: Was oligomerization in both the organic phase and the aqueous (inorganic) phase based on self-dimerization of individual products, or lumped products? In other words, were cross-reactions possible between molecules lumped into a single bin?

**Response:**

In UNIPAR model, oligomerization of organic compounds is processed in both the organic and the inorganic phases based on the self-dimerization reaction of lumped products. The cross-reactions between lumped species are complex. If highly reactive organic species react with weakly reactive species, this oligomerization may be less than the reaction of highly reactive organic species in a single bin but greater than the reaction of weakly reactive organic species in a single bin. Ultimately, they can be compensated in the production of organic matter. Thus, oligomerization in UNIAPR was treated as self-dimerization of lumped products within a single bin (Jang et al., 2005;Jang et al., 2006).

**(3)** p. 10 line 32: The authors identify a large temperature effect on SOA yields, as seen in other studies. However, the authors are uniquely positioned to identify whether this effect is due only to partitioning, as typically assumed, or is also due to temperature dependent reactions that either destroy condensable species or produce other species that hinder gas-to-particle transfers. Partitioning seems to be such a minor SOA source in this study that it is surprising that the observed temperature effects are so pronounced. Could the authors probe the cause of the temperature effect?

**Response:**

Gaseous lumped species first partitions onto aerosols based on a traditional gas-particle partitioning theory. In UNIAPR, the oligomerizations rate is processed by a second order reaction as follows.

$$\frac{dC_{or,i}}{dt} = -k_{o,i} C'^{2}_{or,i} \left( \frac{MW_i OM_T}{\rho_{or} \, 10^3} \right), \qquad \text{(Eq. 6 in the manuscript)}$$

$$\frac{dC_{in,i}}{dt} = -k_{AC,i} C'^{2}_{in,i} \left( \frac{MW_i M_{in}}{\rho_{in} \, 10^3} \right), \qquad \text{(Eq. 7 in the manuscript)}$$

The oligomerization rate constants (L mol⁻¹ s⁻¹) in the organic (*or*) phase and inorganic (*in*) phase are $k_{o,i}$ and $k_{AC,i}$, respectively. The bracketed terms in the equations are the conversion factors from aerosol-base concentrations ($C'_{or,i}$ and $C'_{in,i}$: mol L⁻¹) into air-base concentrations (µg m⁻³) (Section S5). $\rho_{or}$ and $\rho_{in}$ represent the density of the *or* and *in* aerosol. The concentrations (µg m⁻³ of air) of species *i* in the gas phase ($C_{g,i}$), *or* phase ($C_{or,i}$) and *in* phase ($C_{in,i}$) are estimated using partitioning coefficients in the multiphase: i.e., $K_{or,i}$ (m³ µg⁻¹, g /*or*) and $K_{in,i}$ (m³ µg⁻¹, g /*in*). $K_{or,i}$ and $K_{in,i}$ are calculated (by Pankow(Pankow, 1994)) by using estimated activity coefficients in the

*or* and *in* phase ($\gamma_{or,i}$ and $\gamma_{in,i}$), the mass concentration of media ($OM_T$ and $M_{in}$), and vapor pressure ($p_l^\circ$) (See Section 3.3 SOA formation: aerosol-phase reactions). If $C_{or,i}$ and $C_{in,i}$ increase, the formation of oligomerization also increases. Numerous studies have shown that a large fraction of SOA is oligomers (Tolocka et al., 2004;Loeffler et al., 2006;Hoffer, 2004;Baltensperger et al., 2005;Hastings et al., 2005;Riva et al., 2019).

**(4)** In Figure 5, dialkyl organosulfates (diOS) concentrations seem to track ammonium concentrations. In one experiment, the authors comment that diOS formation ceases when the aerosol effloresces. Is there any causative relationship between diOS and ammonium concentrations in wet aerosol?

**Response:**

When acidity is high in wet aerosol, both the diOS formation and the neutralization of acidic sulfate with ammonia can occur. When the inorganic phase is effloresced (no aqueous phase), organic compounds cannot be dissolved in the inorganic phase. Generally, the neutralization of acidic sulfate with ammonia in wet aerosol is faster than OS formation. When the gaseous ammonia concentration is high, sulfuric acid will be consumed by ammonia and thus the formation of diOS can be less. The formation of diOS depends on the concentrations of both acidic sulfate and reactive organic species in the aerosol phase. The prediction of the diOS formation ($[diOS]_{model}$) is performed using the semiempirical equation derived previously for several SOA systems such as aromatic and isoprene SOA (Im et al., 2014;Beardsley and Jang, 2016) as follows,

$$\frac{[diOS]_{model}}{[SO_4^{2-}]_{free}} = 1 - \frac{1}{1 + f_{diOS}\frac{N_{diOS}}{[SO_4^{2-}]_{free}}}, \qquad \text{(Eq. 9 in the manuscript)}$$

$N_{diOS}$ represents the numeric parameter originating from the quantity of reactive chemical species available to form diOS. This parameter is near-explicitly predicted in the model. $f_{diOS}$ represents the semi-empirically determined diOS conversion factor using various chamber data. At each time step, acidic free sulfate ($[SO_4^{2-}]_{free}$), which is the sulfate that is unassociated with ammonium ($[NH_4^+]$), is estimated as ($[SO_4^{2-}] - 0.5\,[NH_4^+]$) and applied to estimate $[diOS]_{model}$. As seen in the equation above, diOS formation ($\frac{[diOS]_{model}}{[SO_4^{2-}]_{free}}$) is not linearly related to $[SO_4^{2-}]_{free}$. Generally, the high proton concentration ($[H^+]$) in the aerosol indicates the high concentration of $[SO_4^{2-}]_{free}$

available for diOS formation. In the morning when humidity is high (>80%), gaseous ammonia concentrations are low because they are condensed on the chamber wall. The ammonia concentrations rise in daytime when humidity is low because ammonia is off-gassing from the chamber wall. Hence, the formation rate of diOS decreases by the two reasons: (1) the consumption of $[SO_4^{2-}]_{free}$ due to diOS formation and (2) the neutralization of $[SO_4^{2-}]_{free}$ with gaseous ammonia as shown in Figure 5 (flat diOS curve in the afternoon).

**(5)** Figure 7 implies that in most experiments, aqueous-phase SOA production is much greater than SOA production via traditional partitioning mechanisms, but the authors don't seem to make this comparison or comment on it. Is it a fair comparison?

**Response:**

The traditional partitioning-based SOA models such as two products model (Odum et al., 1996) or several semivolatile surrogates model (e.g., volatility basis set (VBS)) (Donahue et al., 2006) utilizes semiempirical parameters (e.g., the product stoichiometric coefficient ($\alpha$) and gas-particle partitioning coefficient ($K_p$)) for each HC system under a given $NO_x$ condition. The parameters in traditional partitioning-base models are apparently fit to observed SOA mass. Although the theory facilitates the predicting of SOA mass in the absence of inorganic seed, the SOA mass from the traditional surrogate-based partitioning models is not truly partitioning mass. In UNIAPR, $OM_P$ is predicted solely by the partitioning theory using a near-explicit molecular structure with their activity coefficient and vapor pressure and thus $OM_P$ will be less than the mass prediction using a traditional surrogate-based partitioning models.

**(6)** Figure S1 indicates that ozone is generated too quickly in the model, sometimes by a factor of 2 or 3. Can the authors comment on the implications of this overprediction? Is it related to the "artificial OH radicals" added to the model in certain experiments?

**Response:**

The over-prediction of ozone is not due to an artificial OH radical. As shown in the Figure below, ozone is over-predicted by removing make-up of artificial OH radicals. However, both the addition of artificial OH radicals and overestimation of ozone indicate potential problems in MCM for oxidation of monoalkylbenzenes. The MCM developers and other researchers also reported over-prediction of ozone for aromatic photooxidation (Bloss et al., 2005;Wagner et al., 2003). We have

performed numerous chamber experiments. The overestimation of ozone using MCM appears in monoalkylbenzene series, while a good agreement between predictions and observations is found in xylenes, and trimethylbenzenes, terpenes and isoprene. We propose several explanations for the deviation of predicted ozone formation from the observations.

(a) $RO_2$ chemistry in MCM mechanisms is still uncertain. Numerous products and reactions are involved to form $RO_2$. The cross-reactions between $RO_2$ and the reaction of various $RO_2$ with $HO_2$ are complex (Villenave et al., 1998;Jokinen et al., 2014). Oversimplified surrogate coefficients for the diverse $RO_2$ chemistry could trigger the discrepancy between the modeled and the measured OH radical concentration.

(b) In the gas kinetic mechanism, the photolysis rate constants of organic compounds were also oversimplified using surrogate compounds and can cause uncertainty in ozone prediction and production distributions.

(c) The recent laboratory investigation shows the significance of gas-wall partitioning of organic compounds. The loss of oxygenated products to the chamber wall can lead the lower ozone measurements than the model prediction.

[Figure]

**Minor comments:**

**(1)** p. 2 line 5: Unclear comparison: "… fewer global emissions" that what?

**Response:**

We changed the sentence to "…, despite fewer global emissions compared with biogenic VOCs."

**(2)** p. 2 line 11: When the authors refer to "regional weather" do they really mean climate change?

p. 4 line 9: unclear comparison with literature: What system was "the reported value of 2" measured for? Is it a toluene oxidation study?

**Response:**

p. 2 line 11: We referred to climate change. We changed the sentence to "SOA formation has attracted substantial interest from scholars because of its vital role in affecting climate change, …"

p. 4 line 9: It is measure from a series of toluene oxidation study. We changed the sentence to "The ratio of organic matter (OM) to OC was experimentally determined to be 1.9 (Table 1, EB4), the reported value of 2 in a series of toluene-$NO_x$ oxidation study (Kleindienst et al., 2007)."

**(3)** p. 6 line 18: another unclear comparison with literature: What system was "the reported value of 1.4 g/cm$^3$" measured for?

**Response:**

We changed the sentence to "$\rho_{or}$ and $\rho_{in}$ represent the density of the aerosol of *or* and *in* aerosol. $\rho_{or}$ was experimentally determined (EB4 in Table 1) to be 1.38 g cm$^{-3}$, which was similar to the reported value of 1.4 g/cm$^3$ for aromatic SOA (Ng et al., 2007;Nakao et al., 2011;Chen et al., 2017)."

**(4)** p. 7 line 2: It appears that "irreversibility and nonvolatility" should be "irreversibly formed and nonvolatile".

**Response:**

This has been corrected in the revised manuscript.

**(5)** p. 9 line 12: The meaning of the phrase "contributed to" is unclear in this sentence.

**Response:**

We changed the sentence to "… The formation of aromatic SOAs is attributed to a few highly reactive species, such as GLY…"

**Reference**

Baltensperger, U., Kalberer, M., Dommen, J., Paulsen, D., Alfarra, M. R., Coe, H., Fisseha, R., Gascho, A., Gysel, M., Nyeki, S., Sax, M., Steinbacher, M., Prevot, A. S. H., Sjögren, S., Weingartner, E., and Zenobi, R.: Secondary organic aerosols from anthropogenic and biogenic precursors, Faraday Discuss, 130, 265, 10.1039/b417367h, 2005.

Beardsley, R. L., and Jang, M.: Simulating the SOA formation of isoprene from partitioning and aerosol phase reactions in the presence of inorganics, Atmospheric Chemistry and Physics, 16, 5993-6009, 10.5194/acp-16-5993-2016, 2016.

Bloss, C., Wagner, V., Jenkin, M. E., Volkamer, R., Bloss, W. J., Lee, J. D., Heard, D. E., Wirtz, K., Martin-Reviejo, M., Rea, G., Wenger, J. C., and Pilling, M. J.: Development of a detailed chemical mechanism (MCMv3.1) for the atmospheric oxidation of aromatic hydrocarbons, Atmospheric Chemistry and Physics, 5, 641-664, DOI 10.5194/acp-5-641-2005, 2005.

Chen, L. H., Bao, K. J., Li, K. W., Lv, B., Bao, Z. E., Lin, C., Wu, X. C., Zheng, C. H., Gao, X., and Cen, K. F.: Ozone and Secondary Organic Aerosol Formation of Toluene/NOx Irradiations under Complex Pollution Scenarios, Aerosol Air Qual Res, 17, 1760-1771, 10.4209/aaqr.2017.05.0179, 2017.

Donahue, N. M., Robinson, A. L., Stanier, C. O., and Pandis, S. N.: Coupled partitioning, dilution, and chemical aging of semivolatile organics, Environmental Science & Technology, 40, 2635-2643, 10.1021/es052297c, 2006.

Hastings, W. P., Koehler, C. A., Bailey, E. L., and De Haan, D. O.: Secondary organic aerosol formation by glyoxal hydration and oligomer formation: Humidity effects and equilibrium shifts during analysis, Environmental Science & Technology, 39, 8728-8735, 2005.

Hoffer, A.: Chemical characterization of humic-like substances (HULIS) formed from a lignin-type precursor in model cloud water, Geophys Res Lett, 31, 10.1029/2003gl018962, 2004.

Im, Y., Jang, M., and Beardsley, R. L.: Simulation of aromatic SOA formation using the lumping model integrated with explicit gas-phase kinetic mechanisms and aerosol-phase reactions, Atmospheric Chemistry and Physics, 14, 4013-4027, 10.5194/acp-14-4013-2014, 2014.

Jang, M., Czoschke, N. M., Northcross, A. L., Cao, G., and Shaof, D.: SOA formation from partitioning and heterogeneous reactions: Model study in the presence of inorganic species, Environmental Science & Technology, 40, 3013-3022, 10.1021/es0511220, 2006.

Jang, M. S., Czoschke, N. M., and Northcross, A. L.: Semiempirical model for organic aerosol growth by acid-catalyzed heterogeneous reactions of organic carbonyls, Environmental Science & Technology, 39, 164-174, 10.1021/es048977h, 2005.

Jokinen, T., Sipilä, M., Richters, S., Kerminen, V.-M., Paasonen, P., Stratmann, F., Worsnop, D., Kulmala, M., Ehn, M., Herrmann, H., and Berndt, T.: Rapid Autoxidation Forms Highly Oxidized RO 2 Radicals in the Atmosphere, 53, 14596-14600, 10.1002/anie.201408566, 2014.

Kleindienst, T. E., Jaoui, M., Lewandowski, M., Offenberg, J. H., Lewis, C. W., Bhave, P. V., and Edney, E. O.: Estimates of the contributions of biogenic and anthropogenic hydrocarbons to secondary organic aerosol at a southeastern US location, Atmospheric Environment, 41, 8288-8300, 10.1016/j.atmosenv.2007.06.045, 2007.

Loeffler, K. W., Koehler, C. A., Paul, N. M., and De Haan, D. O.: Oligomer formation in evaporating aqueous glyoxal and methyl glyoxal solutions, Environmental Science & Technology, 40, 6318-6323, 2006.

Nakao, S., Clark, C., Tang, P., Sato, K., and Cocker, D.: Secondary organic aerosol formation from phenolic compounds in the absence of NOx, Atmospheric Chemistry and Physics, 11, 10649-10660, 10.5194/acp-11-10649-2011, 2011.

Ng, N. L., Kroll, J. H., Chan, A. W. H., Chhabra, P. S., Flagan, R. C., and Seinfeld, J. H.: Secondary organic aerosol formation from m-xylene, toluene, and benzene, Atmospheric Chemistry and Physics, 7, 3909-3922, DOI 10.5194/acp-7-3909-2007, 2007.

Odum, J. R., Hoffmann, T., Bowman, F., Collins, D., Flagan, R. C., and Seinfeld, J. H.: Gas/particle partitioning and secondary organic aerosol yields, Environmental Science & Technology, 30, 2580-2585, Doi 10.1021/Es950943+, 1996.

Pankow, J. F.: An absorption model of the gas/aerosol partitioning involved in the formation of secondary organic aerosol, Atmos. Environ., 28, 189-193, 1994.

Riva, M., Heikkinen, L., Bell, D. M., Peräkylä, O., Zha, Q., Schallhart, S., Rissanen, M. P., Imre, D., Petäjä, T., Thornton, J. A., Zelenyuk, A., and Ehn, M.: Chemical transformations in monoterpene-derived organic aerosol enhanced by inorganic composition, npj Climate and Atmospheric Science, 2, 10.1038/s41612-018-0058-0, 2019.

Tolocka, M. P., Jang, M., Ginter, J. M., Cox, F. J., Kamens, R. M., and Johnston, M. V.: Formation of Oligomers in Secondary Organic Aerosol, Environmental Science & Technology, 38, 1428-1434, 10.1021/es035030r, 2004.

Villenave, E., Lesclaux, R., Seefeld, S., and Stockwell, W. R.: Kinetics and atmospheric implications of peroxy radical cross reactions involving the CH3C(O)O2radical, Journal of Geophysical Research: Atmospheres, 103, 25273-25285, 10.1029/98jd00926, 1998.

Wagner, V., Jenkin, M. E., Saunders, S. M., Stanton, J., Wirtz, K., and Pilling, M. J.: Modelling of the photooxidation of toluene: conceptual ideas for validating detailed mechanisms, Atmospheric Chemistry and Physics, 3, 89-106, 10.5194/acp-3-89-2003, 2003.